# The non-vesicular cholesterol transporter GRAMD1C is a pan-coronavirus antiviral target

Zhelin Su[1,2], Zhen Fu[1,2], Yichen Yang[1,2], Yanan Fu[1,2], Jingwen Mo[3], Juan Xu[1,2], Yubei Tan[1,2], Yixin Xiang[1,2], Yuejun Shi[1,2], Shengsong Xie[4], Limeng Sun[1,2]*, Guiqing Peng[1,2,5]*

1 State Key Laboratory of Agricultural Microbiology, College of Veterinary Medicine, Huazhong Agricultural University, Wuhan, Hubei, P. R. China, 2 Key Laboratory of Preventive Veterinary Medicine in Hubei Province, The Cooperative Innovation Center for Sustainable Pig Production, Wuhan, Hubei, P. R. China, 3 University of Science and Technology of China, Hefei, Anhui, P. R. China, 4 Key Laboratory of Agricultural Animal Genetics, Breeding and Reproduction of Ministry of Education & Key Lab of Swine Genetics and Breeding of Ministry of Agriculture and Rural Affairs, Huazhong Agricultural University, Wuhan, Hubei, P. R. China, 5 State Key Laboratory of Agricultural Microbiology, Key Laboratory of Prevention & Control for African Swine Fever and Other Major Pig Diseases, Ministry of Agriculture and Rural Affairs, College of Veterinary Medicine, Huazhong Agricultural University, Wuhan, Hubei, P. R. China

* sunlimeng@webmail.hzau.edu.cn (LS); penggq@mail.hzau.edu.cn (GP)

## Abstract

Coronaviruses (CoVs) rely heavily on host lipid metabolism for efficient replication, but the specific host pathways involved remain incompletely defined. Here, we identify the non-vesicular cholesterol transport protein GRAMD1C as a key host factor required for the replication of multiple coronaviruses across α, β, and δ genera, including TGEV, PEDV, HCoV-229E, SARS-CoV-2, MHV, and PDCoV. Mechanistically, GRAMD1C is recruited to the replication factory through its interaction with the viral nonstructural proteins 3 and 4 (nsp3 and nsp4). Transmission electron microscopy (TEM) analysis revealed that GRAMD1C facilitates the formation of replication double-membrane vesicles (DMVs). Further domain rescue and inhibitor experiments demonstrated that GRAMD1C's cholesterol transport activity is essential for viral replication. Moreover, GRAMD1C-deficient and inhibitor-treated mice exhibited reduced viral replication, underscoring its critical role in vivo. Collectively, our findings expand the current understanding of the importance of non-vesicular cholesterol transport in viral replication and highlight GRAMD1C as a promising broad-spectrum antiviral target.

## Introduction

Coronaviruses (CoVs) are enveloped, positive-sense, single-stranded RNA viruses classified into the α, β, γ, and δ genera [1,2]. These viruses have caused severe global public health emergencies, notably severe acute respiratory syndrome coronavirus 2 (SARS-CoV-2) [3–6]. Animal CoVs such as porcine transmissible gastroenteritis virus (TGEV) and murine hepatitis virus (MHV) share key genomic structural

**Data availability statement:** All relevant data are within the paper and its Supporting information files.

**Funding:** This work was supported by the National Science Fund for Distinguished Young Scholars (32125037 to GP) (http://www.nsfc.gov.cn/), the China Postdoctoral Science Foundation (2023M731233 to LS) (https://www.chinapostdoctor.org.cn/), the National Natural Science Foundation of China (32202784 to LS) (https://www.nsfc.gov.cn/) and the Science Fund for Creative Research Groups of the Natural Science Foundation of Hubei Province (2022CFA027 to GP) (http://kjt.hubei.gov.cn/). The funders had no role in study design, data collection and analysis, decision to publish, or preparation of the manuscript. No authors received a salary from any funders.

**Competing interests:** The authors have declared that no competing interests exist.

**Abbreviations:** 20α-HC, 20α-hydroxycholesterol; cDNAs, complementary DNAs; Co-IP, co-immunoprecipitation; CoVs, coronaviruses; DAPI, 4′,6-diamidino-2-phenylindole; DMEM, Dulbecco's Modified Eagle's Medium; DMVs, double-membrane vesicles; dsRNA, double-stranded RNA; ER, endoplasmic reticulum; FBS, fetal bovine serum; GRAMD1C, GRAM domain-containing 1C; HCV, hepatitis C virus; H&E, Hematoxylin and eosin; IFA, immunofluorescence assay; IHC, immunohistochemistry; KO, knockout; LPDS, lipoprotein deficient serum; MHV, murine hepatitis virus; MOI, multiplicities of infection; NPC1, Niemann-Pick type C protein 1; OSBP, oxysterol-binding protein; PBS, phosphate-buffered saline; PEDV, porcine epidemic diarrhea virus; PM, plasma membrane; ROI, region of interest; RTCs, replication-transcription complexes; RT-qPCR, real-time quantitative PCR; SARS-CoV-2, severe acute respiratory syndrome coronavirus 2; SD, standard deviation; TEM, transmission electron microscopy; TGEV, transmissible gastroenteritis virus; TM, C-terminal transmembrane; WT, wild-type.

features and replication-transcription mechanisms with SARS-CoV-2. This conservation establishes TGEV and MHV as valuable models for studying coronavirus pathogenesis and developing antiviral drugs and vaccines [7,8].

Coronavirus replication involves multiple key steps: viral entry, uncoating and translation of the viral RNA, assembly of the replication-transcription complexes (RTCs), viral RNA replication and transcription, and viral particle assembly and release [9–12]. Coronaviruses establish their replication machinery within specialized double-membrane vesicles (DMVs). These DMVs, formed by remodeling the host endoplasmic reticulum (ER), serve as shielded compartments for viral RNA synthesis and protection against host immune surveillance [13–15]. During authentic infection, viral nonstructural proteins nsp3, nsp4, and nsp6 collectively contribute to DMV biogenesis [16–19]. Among these, co-expression of nsp3 and nsp4 has been established as the minimal requirement sufficient to induce DMV formation in vitro [20,21], highlighting their central role in initiating membrane remodeling. The role of cholesterol in the formation of replication organelles has been well-demonstrated for other viruses, such as hepatitis C virus (HCV) [22,23] and Enteroviruses [24]. As a key lipid component of ER membranes, cholesterol enriches at DMV assembly sites to maintain membrane fluidity and curvature. Pharmacological targeting of cholesterol disrupts both disease progression and viral replication cycles, positioning cholesterol metabolism as a pivotal host-directed target against viral pathogenesis [25–27].

Recent studies utilizing CRISPR screening have identified several cholesterol-associated host factors in viral replication, including Niemann-Pick type C protein 1 (NPC1) and Sterol Regulatory Element-Binding Transcription Factor 2 (SREBF2) in coronaviruses [28,29] and Apolipoprotein E (APOE) in influenza virus [30]. Our prior genome-scale CRISPR screen identified GRAM domain-containing 1C (GRAMD1C) as a host factor associated with coronavirus replication [31]. This ER-localized protein mediates non-vesicular cholesterol transport. When cholesterol levels in the plasma membrane (PM) rise above a certain threshold, GRAMD1s proteins are dynamically recruited to PM-ER contact sites where they bind to cholesterol and facilitate its non-vesicular transport to the ER [32]. Importantly, no prior study has implicated GRAMD1C in viral propagation.

Here, using TGEV as a model, we found that GRAMD1C knockout (KO) significantly inhibited CoVs replication. Mechanistically, GRAMD1C is recruited to RTCs through its interaction with nsp3 and nsp4, enabling cholesterol flux necessary for DMV morphogenesis. GRAMD1C inhibition confers broad-spectrum suppression of coronavirus replication. In animal experiments, heterozygous GRAMD1C KO (GRAMD1C$^{+/-}$) mice and inhibitor-treated mice exhibited superior antiviral defense. In conclusion, our work identifies GRAMD1C as a therapeutic target and elucidates its role in CoVs replication.

## Results

### GRAMD1C is a pan-host factor required for coronavirus replication

To validate the role of GRAMD1C in the replication of the α coronavirus TGEV, we generated a stable GRAMD1C-KO PK-15 cell line using the CRISPR/Cas9

gene-editing system. Genomic sequence analysis confirmed a frameshift mutation characterized by an 8-nucleotide deletion in the GRAMD1C-KO cells (Fig 1A). Western blot analysis further confirmed the loss of GRAMD1C protein in these cells (Fig 1B). No significant cytotoxicity was observed in the GRAMD1C-KO cells, as demonstrated by the 5-ethynyl-2′-deoxyuridine (EdU) fluorescence assay and the (3-(4,5-dimethylthiazol-2-yl)-5-(3-carboxymethoxyphenyl)-2-(4-sulfophenyl)-2H tetrazolium) (MTS) assay (S1A and S1B Fig).

The absence of GRAMD1C led to a marked inhibition of TGEV replication, as evidenced by reduced expression levels of the TGEV-encoded nucleocapsid (N) protein by western blot and real-time quantitative PCR (RT-qPCR) assays (Fig 1C and 1D). To further assess the impact of GRAMD1C on virus replication, we quantified viral loads at indicated time points (Fig 1E) and performed immunofluorescence assays (IFA) under various multiplicities of infection (MOI) (Fig 1F). The results clearly demonstrated that GRAMD1C depletion significantly reduced TGEV viral load. To investigate whether overexpression of GRAMD1C could enhance virus infectivity, we overexpressed GRAMD1C in wild-type (WT) cells. RT-qPCR and viral titer assays revealed that TGEV replication was promoted by GRAMD1C overexpression (Fig 1G). Together, these data suggest that GRAMD1C is a critical host factor for TGEV infection.

To determine whether GRAMD1C is required for the replication of other coronaviruses, GRAMD1C-KO Vero-E6 cells were constructed (S1C and S1D Fig) and infected with porcine epidemic diarrhea virus (PEDV). IFA and viral titer assays confirmed a significant decrease in PEDV replication in the KO cells (Fig 1H). For other alphacoronaviruses such as HCoV-229E, we conducted IFA and viral titer assays in Caco-2 cells (Figs 1I, S1E, and S1F). The results indicated that GRAMD1C KO significantly suppressed the replication of HCoV-229E. As for betacoronaviruses, we investigated the replication of the original SARS-CoV-2 strain in GRAMD1C-KO Vero-E6 cells. IFA, viral titer assays, and western blot analysis all revealed a decrease in viral replication compared to WT cells (Fig 1J). Similarly, GRAMD1C-KO clones in murine fibroblast L929 cells exhibited significantly lower viral titers of murine hepatitis virus (MHV-A59) compared to WT L929 cells (Figs 1K, S1G, and S1H). To explore the impact of GRAMD1C on porcine deltacoronavirus (PDCoV) replication, we infected GRAMD1C-KO PK-15 cells with PDCoV. IFA and viral titer assays showed a pronounced reduction in PDCoV replication in the KO cells (Fig 1L).

## GRAMD1C modulates viral replication via DMV formation

Given that both GRAMD1C and viral replication organelles are associated with the ER, we first examined whether GRAMD1C affects the viral replication stage. However, we observed a decrease in the expression of the TGEV receptor ANPEP in GRAMD1C-KO PK-15 cells (S2A Fig). A similar downregulation of *ANPEP* was also detected in GRAMD1C-KO Caco-2 cells (S2B Fig). To determine whether GRAMD1C's effect on viral receptors is specific to ANPEP, we measured the expression levels of *ACE2* and *CEACAM1* in GRAMD1C-KO Vero-E6 and GRAMD1C-KO L929 cells, respectively. RT-qPCR analysis revealed a decrease in *ACE2* expression, whereas *CEACAM1* levels remained unchanged (S2C and S2D Fig). Therefore, to directly examine the replication stage while bypassing the viral entry process, we generated ANPEP-KO PK-15 and ANPEP-GRAMD1C double-KO (ANPEP-GRAMD1C-KO) PK-15 cell lines (S3A and S3B Fig) and confirmed that the transfection efficiencies of the two cell lines were comparable (S4A and S4B Fig). We then transfected these cells with a TGEV bacterial artificial chromosome (BAC) plasmid containing the viral genomic cDNA. Within the transfected cells, infectious progeny virions were assembled and released (Fig 2A) [33]. RT-qPCR assays showed that viral replication was further reduced in ANPEP-GRAMD1C-KO cells compared to ANPEP-KO cells (Fig 2B), confirming that GRAMD1C affects intracellular viral replication independently of viral entry.

We also reconstituted ANPEP expression in the GRAMD1C-KO PK-15 cells (GRAMD1C-KO+ANPEP), restoring ANPEP expression to WT levels. (S2A Fig). Attachment and internalization assays confirmed that viral entry was restored to WT levels upon ANPEP complementation (S5A and S5B Fig). Under these conditions, TGEV-BAC transfection in WT and GRAMD1C-KO+ANPEP cells revealed that GRAMD1C deficiency still significantly impaired viral replication (Fig 2C). Consistent with this, infection of GRAMD1C-KO+ANPEP cells with live virus, followed by RT-qPCR, multi-time point viral

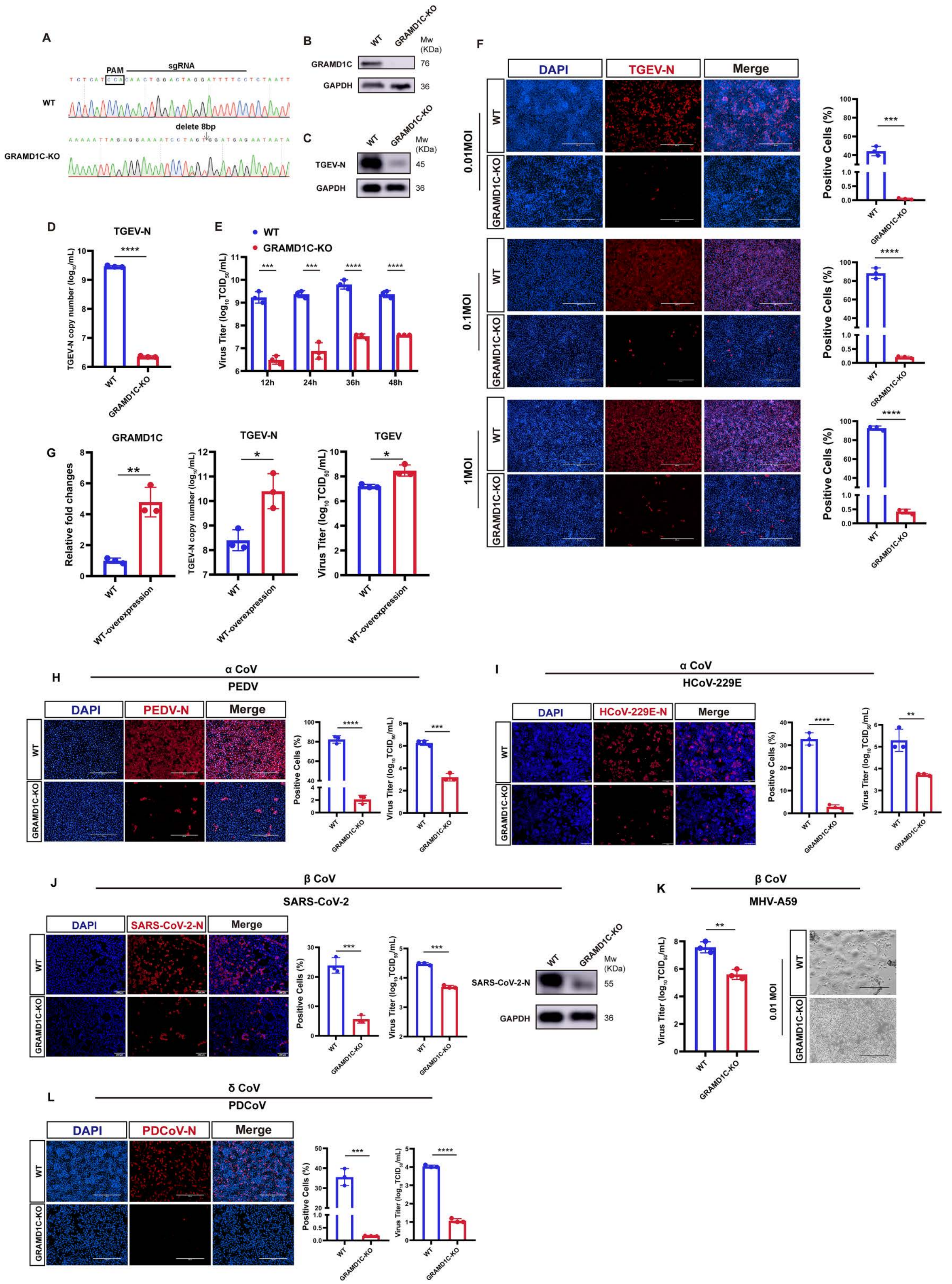


**Fig 1. GRAMD1C is a pan-host factor required for coronaviruses replication. (A)** Genomic sequence analysis of *GRAMD1C* in WT cells and GRAMD1C-KO PK-15 cells. **(B)** western blot analysis validated the protein expression level of endogenous GRAMD1C in WT cells and KO cells. **(C)** GRAMD1C-KO and WT cells were infected with TGEV (MOI = 0.1) and harvested at 24 hpi for western blot analysis. GAPDH used as an internal control

gene. **(D)** The copy number of the TGEV N gene was determined in GRAMD1C-KO and WT cells at 24 hpi, following infection with TGEV (MOI = 0.1), using absolute quantitative real-time PCR. **(E)** Multiple-step viral growth assay. The viral titers in GRAMD1C-KO and WT cells infected with TGEV (MOI = 1) were monitored at different time points (12, 24, 36, and 48 h). **(F)** GRAMD1C-KO and WT cells were infected with TGEV at different MOIs (0.01, 0.1, and 1). Immunofluorescence assays were used to detect the TGEV N protein expression at 24 hpi. Scale bar, 400 μm. **(G)** Overexpression of GRAMD1C in PK-15 control cells at 24 hpi post-TGEV infection (MOI = 0.01). RT-qPCR and viral titer assays for determination of the TGEV replication. WT, transfection of the pCAGGS-HA empty vector in PK-15 cells; WT-overexpression, transfection of the pCAGGS-GRAMD1C-HA vector in PK-15 cells. **(H)** WT and GRAMD1C-KO Vero-E6 cells were infected with PEDV at 0.1 MOI. Immunofluorescence assays were used to detect the PEDV N protein expression at 24 hpi. Scale bar, 400 μm. The virus titers were tested at 24 hpi. **(I)** WT and GRAMD1C-KO Caco-2 cells were infected with HCoV-229E at 0.1 MOI. Immunofluorescence assays were used to detect the HCoV-229E N protein expression at 36 hpi. Scale bar, 200 μm. The virus titers were tested at 36 hpi. **(J)** WT and GRAMD1C-KO Vero-E6 cells were infected with SARS-CoV-2 at 0.01 MOI. Immunofluorescence assays were used to detect the SARS-CoV-2 N protein expression at 24 hpi. Scale bar, 200 μm. The virus titers were tested at 24 hpi. Western blot analysis for SARS-CoV-2 N protein at 24 hpi (MOI = 0.1). GAPDH used as an internal control gene. **(K)** WT and GRAMD1C-KO L929 cells were infected with MHV (MOI = 0.01). MHV titers were tested at 12 hpi. WT cells exhibited pronounced cytopathic effects compared to KO cells. **(L)** WT and GRAMD1C-KO PK-15 cells were infected with PDCoV at 5 MOI in the presence of 7.5 μg/ml trypsin. Immunofluorescence assays were used to detect the PDCoV N protein expression at 24 hpi. Scale bar, 400 μm. PDCoV titers were tested at 24 hpi. The means and SDs of the results from three independent experiments are shown. $*P < 0.05$; $**P < 0.01$; $***P < 0.001$; $****P < 0.0001$. $P$-values were determined by two-tailed Student's $t$-tests. The data underlying this Figure can be found in S1 Data and S1 Raw Images.

titer assays, and IFA, showed that the antiviral phenotype persisted and was comparable to that in GRAMD1C-KO cells (Fig 2D–2F).

To probe the mechanistic basis of this replication defect, we performed confocal microscopy assay at 3 h post-infection (hpi), which revealed a significant decrease in the formation of viral double-stranded RNA (dsRNA), a replication intermediate, in GRAMD1C-KO+ANPEP cells compared with WT cells (Fig 2G). Transmission electron microscopy (TEM) at 12 hpi to visualize replication organelles showed that DMVs were scarcely detected in GRAMD1C-KO+ANPEP cells (Fig 2H). To test if GRAMD1C is directly required for DMV biogenesis, we co-transfected WT and GRAMD1C-KO cells with TGEV nsp3 and nsp4, which are sufficient to form viral DMVs [20,21]. The results showed that DMVs readily formed adjacent to the ER in WT cells, but were almost absent in GRAMD1C-KO cells (Fig 2I), indicating that the loss of GRAMD1C directly impairs DMV formation.

Collectively, these results indicate that GRAMD1C is critical for DMV formation and viral replication.

## The GRAMD1C transmembrane domain is required for interaction with DMV-associated viral nonstructural proteins and viral replication

To investigate whether GRAMD1C interacts with DMV-associated proteins, we co-expressed GRAMD1C, nsp3, and nsp4 in HEK-293T cells and analyzed their subcellular localization. To better visualize the subcellular localization of GRAMD1C, we exogenously overexpressed a FLAG-tagged GRAMD1C, instead of detecting the endogenous protein. When nsp3 or nsp4 was expressed individually, GRAMD1C colocalized with them on the ER. In cells co-expressing nsp3/4, there was a strong colocalization of GRAMD1C with the puncta labeled by nsp3/4 (Fig 3A). Pearson's correlation analysis revealed a high degree of co-localization between GRAMD1C and the viral proteins, with coefficients of 0.992 for nsp4 and 0.988 for nsp3. Notably, we observed that GRAMD1C consistently co-localized with TGEV-nsp3/4 in punctate structures under both lipid-poor (1% lipoprotein-deficient serum, LPDS) and cholesterol-loaded conditions, indicating its retention on DMV membranes following their formation (S6A and S6B Fig). Co-immunoprecipitation (Co-IP) assays confirmed an interaction between GRAMD1C and TGEV-nsp4 (Fig 3B). An interaction with nsp3 was also detected (S7A Fig). Together with the colocalization analysis, these results suggest that GRAMD1C may bind to nsp3 and nsp4 with comparable affinity. However, due to the large size of nsp3 which posed technical challenges, subsequent investigations were primarily focused on nsp4.

GRAMD1C contains an N-terminal GRAM domain, a central ASTER domain, and a C-terminal transmembrane (TM) domain. The TM domain anchors GRAMD1C to the ER, while the GRAM and ASTER domains mediate its localization to PM–ER junctions and facilitate cholesterol transfer under high membrane cholesterol conditions [34]. To identify the

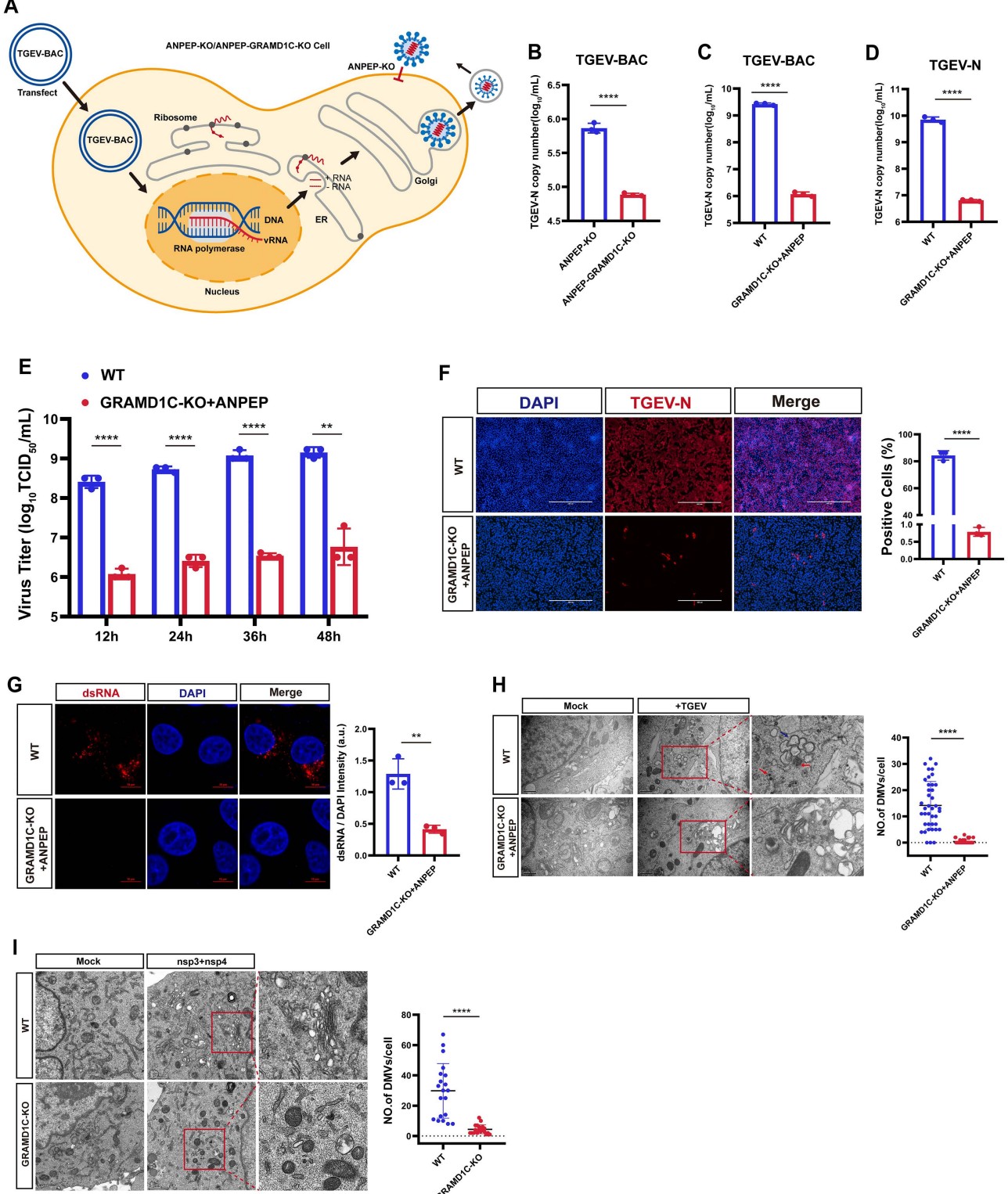

**Fig 2. GRAMD1C modulates viral replication via DMV formation. (A)** A diagram of cells transfected with TGEV-BAC plasmids. Virions bypass the entry phase and are generated intracellularly. **(B)** ANPEP-KO and ANPEP-GRAMD1C-KO cells were transfected with TGEV-BAC plasmids. Cells were harvested at 60 hours post-transfection and analyzed by RT-qPCR. **(C)** WT and GRAMD1C-KO+ANPEP cells were transfected with TGEV-BAC

plasmids. Cells were harvested at 60 hours post-transfection and analyzed by RT-qPCR. **(D)** The copy number of the TGEV N gene was determined in WT and GRAMD1C-KO+ANPEP cells at 24 hpi, following infection with TGEV (MOI = 0.1), using absolute quantitative real-time PCR. **(E)** The viral titers in WT and GRAMD1C-KO+ANPEP cells infected with TGEV (MOI = 0.1) were monitored at different time points (12, 24, 36, and 48 h). **(F)** WT and GRAMD1C-KO+ANPEP cells were infected with TGEV at 0.1 MOI. Immunofluorescence assays were used to detect the TGEV N protein expression at 24 hpi. Scale bar, 400 µm. **(G)** Confocal microscopy to evaluate early-stage TGEV replication by detecting dsRNA formation in WT and GRAMD1C-KO+ANPEP cells at 3 hpi after infection with TGEV (MOI = 5). Scale bars, 10 µm. **(H)** Evaluation of the effects of GRAMD1C-KO+ANPEP cells on virus particle assembly by TEM experiment. Compared with numerous virions (red arrows) and typical DMVs (blue arrows) found in WT cells, few virus-like particles wrapped in vesicles of varying sizes were found in GRAMD1C-KO+ANPEP cells. Scale bars, 1 µm or 0.5 µm. **(I)** WT and GRAMD1C-KO cells transfected with TGEV-nsp3/nsp4 plasmids were analyzed by TEM to assess viral replication organelle formation. Representative images show DMVs at 24 hours post-transfection. DMVs were manually counted within complete single-cell profiles (*n* = 20 cell profiles per group). Scale bar, 1 µm or 0.5 µm. The means and SDs of the results from three independent experiments are shown. ns, not significant; **$P < 0.01$; ****$P < 0.0001$. *P*-values were determined by two-tailed Student's *t*-tests. The data underlying this Figure can be found in S2 Data.

GRAMD1C–nsp4 interaction interface, molecular docking was performed to model their binding. Key interacting residues in GRAMD1C were predicted to be F117, I559, L566, L595, L642, and F649, located primarily within the transmembrane and luminal domains (Fig 3C). Each of these residues was individually mutated to alanine (A). However, single-point mutations were insufficient to abolish the interaction between GRAMD1C and TGEV-nsp4 (S7B and S7C Fig). We therefore constructed a truncation mutant lacking the TM and luminal domains of GRAMD1C (ΔTM-GRAMD1C) (Fig 3C). We co-expressed ΔTM-GRAMD1C, nsp3, and nsp4 and analyzed their subcellular localization. Strikingly, ΔTM-GRAMD1C exhibited a distinct aggregation pattern and failed to co-localize with nsp3/4 (Fig 3D). Co-IP assays revealed that ΔTM-GRAMD1C lost its ability to bind TGEV-nsp4 (Fig 3E).

To assess whether this mechanism is conserved in β-coronaviruses, we examined the interaction between human GRAMD1C and SARS-CoV-2 nsp4. Co-IP confirmed their interaction (Fig 3F), and molecular docking similarly mapped the interface to the TM and luminal domains (Fig 3G). Truncation of these domains abolished binding (Fig 3H). Functional complementation in GRAMD1C-KO PK-15 cells demonstrated that ΔTM-GRAMD1C could not rescue viral replication (Figs 3I and S8), underscoring the essential role of TM-mediated interaction.

Collectively, these results demonstrate that the TM domain of GRAMD1C is required for its association with coronavirus nsp3/4 complexes and for promoting DMV biogenesis and viral replication.

## GRAMD1C-mediated cholesterol transport is critical for viral replication

Having identified the TM domain as the essential binding domain for nsp4, we next investigated how this interaction functionally contributes to the biogenesis of viral replication organelles. GRAMD1C transports cholesterol from the PM to the ER (Fig 4A). Given that the ER harbors viral replication complexes, we hypothesized that cholesterol mobilized by GRAMD1C to the ER might be utilized for the generation of DMVs during viral replication.

To test this hypothesis, we performed viral challenge assays in GRAMD1C-overexpressing cells. Confocal microscopy revealed that GRAMD1C, an ER-resident protein, was recruited to viral dsRNA-positive foci and reorganized into clustered punctate aggregates adjacent to viral replication complexes (Fig 4B). This spatial redistribution suggests GRAMD1C's functional involvement in coronavirus genome replication. Next, to investigate the mechanism of GRAMD1C-dependent cholesterol transport during viral replication, we visualized cholesterol distribution in relation to viral replication sites and ER membrane architecture. PK-15 cells transfected with an ER marker (pDsRed2-ER) to label the ER, were infected with TGEV, and then stained for cholesterol (CY5-cholesterol) and viral dsRNA. Confocal imaging revealed specific, intense accumulation of cholesterol at dsRNA-positive puncta, whereas the ER marker did not cluster at these sites (Figs 4C and S9). Quantitative colocalization analysis confirmed a strong spatial correlation between cholesterol-enriched domains and viral dsRNA, but not between these domains and ER. This indicates that cholesterol enrichment at replication organelles results from a targeted recruitment event, distinct from passive accumulation due to ER membrane remodeling. Subsequently, we monitored changes in cellular free cholesterol dynamics before and after infection using

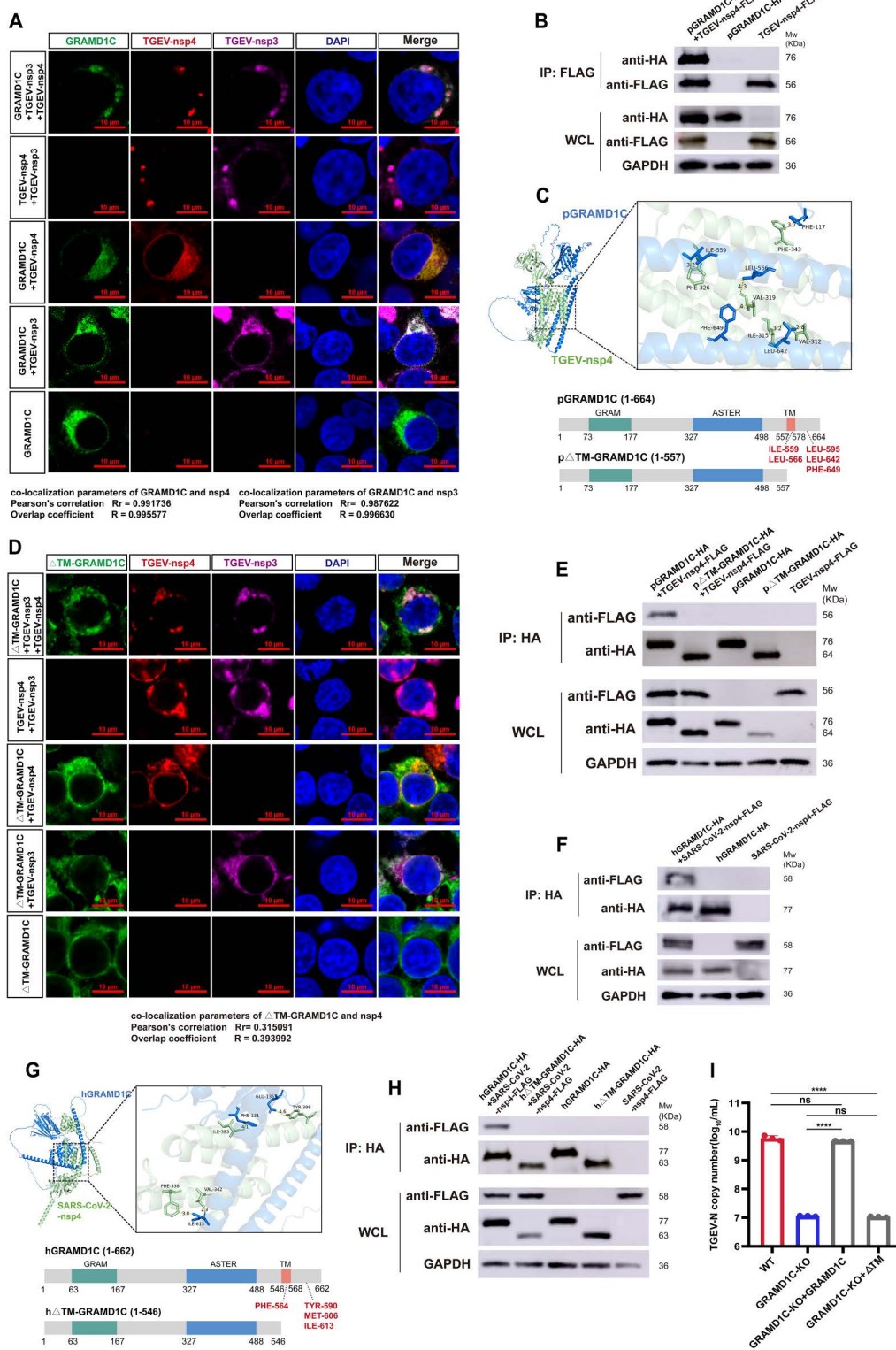

**Fig 3. The GRAMD1C transmembrane domain is required for interaction with DMV-associated nsp4 and viral replication. (A)** Confocal fluorescence microscopy analysis of the co-localization of TGEV-nsp4-His (indicated in red), TGEV-nsp3-HA (indicated in magenta), GRAMD1C-FLAG (indicated in green) in HEK-293Tcells. Scale bar, 10 µm. The co-localization correlation was analyzed. **(B)** Co-IP assay was performed in HEK293T cells transfected with pCAGGS-GRAMD1C-HA and pCAGGS-TGEV-nsp4-FLAG for 24 hours. **(C)** Schematic representation of the simulated protein-protein

interaction modeling. Schematic representation of the construction of full-length GRAMD1C and GRAMD1C lacking the TM and luminal domains (ΔTM). The N-terminal GRAM (green), central ASTER (blue), and transmembrane (TM, red) domains are indicated. Potential interaction sites are annotated in red. **(D)** Confocal fluorescence microscopy analysis of the localization of TGEV-nsp4-His (indicated in red), TGEV-nsp3-HA (indicated in magenta), ΔTM-GRAMD1C-FLAG (indicated in green) in HEK-293Tcells. Scale bar, 10 μm. The co-localization correlation was analyzed. **(E)** Co-IP assay was performed in HEK293T cells transfected with pCAGGS-ΔTM-GRAMD1C-HA and pCAGGS-TGEV-nsp4-FLAG for 24 hours. **(F)** Co-IP assay was performed in HEK293T cells transfected with pCAGGS-human-GRAMD1C-HA and pCAGGS-SARS-CoV-2-nsp4-FLAG for 24 hours. **(G)** Schematic representation of the simulated protein-protein interaction modeling. Schematic representation of the construction of full-length human GRAMD1C and human GRAMD1C lacking the TM and luminal domains (ΔTM). Potential interaction sites are annotated in red. **(H)** Co-IP assay was performed in HEK293T cells transfected with pCAGGS-human-ΔTM-GRAMD1C-HA and pCAGGS-SARS-CoV-2-nsp4-FLAG for 24 hours. **(I)** RT-qPCR assay for determination of TGEV replication in GRAMD1C-KO cells and cells rescued with different domains. Cells were infected with TGEV (MOI = 0.01) and harvested at 24 hpi for RT-qPCR. The means and SDs of the results from three independent experiments are shown. ns, not significant; ****$P < 0.0001$. $P$-values were determined by two-tailed unpaired Student's *t*-tests. The data underlying this Figure can be found in S3 Data and S1 Raw Images.

Filipin III staining. Fluorescence imaging revealed that cholesterol signals were diminished in some PMs and intracellular compartments of PK-15 cells after TGEV infection (Fig 4D), suggesting possible cholesterol consumption during viral replication. To more directly assess alterations in intracellular cholesterol levels upon infection, we performed a biochemical measurement of free cholesterol using a Free Cholesterol Content Assay Kit. The results showed that viral infection led to a significant reduction in intracellular free cholesterol at 24 hpi (Fig 4E), indicating that cholesterol consumption is necessary for productive viral replication.

To further determine the essential role of GRAMD1C-mediated cholesterol transport, we overexpressed various truncations including full-length GRAMD1C, GRAMD1C lacking the GRAM domain (ΔGRAM), and GRAMD1C containing only the GRAM domain (GRAM) in GRAMD1C-KO PK-15 cells (Fig 4F). Western blot confirmed the successful overexpression of these domains (S10 Fig). When full-length GRAMD1C was overexpressed, the viral replication significantly increased compared to the GRAMD1C-KO cells. However, overexpression of ΔGRAM or the GRAM domain alone failed to restore normal viral replication (Fig 4G), indicating that GRAMD1C-mediated cholesterol transport to the RTC is crucial for viral replication.

To further demonstrate that cholesterol is responsible for the observed phenotype, we exogenously supplemented cholesterol in GRAMD1C-KO PK-15 cells to assess whether viral replication could be restored. IFA results showed that exogenous cholesterol supplementation indeed enhanced viral replication in WT PK-15 cells; however, it failed to restore viral replication in GRAMD1C-KO cells (Fig 4H). We propose that this is because GRAMD1C is required for the proper trafficking of cholesterol to sites of DMV biogenesis. In its absence, simply increasing cholesterol availability is insufficient.

## GRAMD1C inhibitor 20α-HC blocks viral DMV biogenesis

Previous research identified that 20α-hydroxycholesterol (20α-HC) acts as a GRAMD1C inhibitor by competing with cholesterol for binding to a hydrophobic pocket within the ASTER domain, thereby blocking GRAMD1C-mediated non-vesicular cholesterol transport [35] (Fig 5A). To further examine the requirement of GRAMD1C-mediated cholesterol transport for viral replication, we treated PK-15 cells with 20α-HC at well-tolerated concentrations (S11A Fig). Treatment with 20α-HC led to a decrease in viral titers and inhibition of TGEV-N protein expression in a dose-dependent manner (Fig 5B and 5C). Akin to GRAMD1C deficiency, 20α-HC treatment downregulated ANPEP expression following 24-hour exposure (S11B Fig). Consequently, to identify whether 20α-HC blocked virus replication, the inhibitor was applied after virus entry and restricted to a 12-hour exposure period. Viral titer assays and RT-qPCR analysis revealed that 20α-HC potently inhibited viral replication (Fig 5D), consistent with the results in GRAMD1C-KO cells. Confocal imaging revealed a marked reduction in viral dsRNA following inhibitor treatment, implying possible impairment of viral replication sites (Fig 5E).

To investigate whether the inhibitor further impaired viral DMV biogenesis, we treated cells co-expressing nsp3/4 with 20α-HC and examined their subcellular localization. Quantitative colocalization analysis showed reduced nsp3/4 co-localization in inhibitor-treated cells (Fig 5F), suggesting potential disruption of DMV biogenesis. For direct visualization

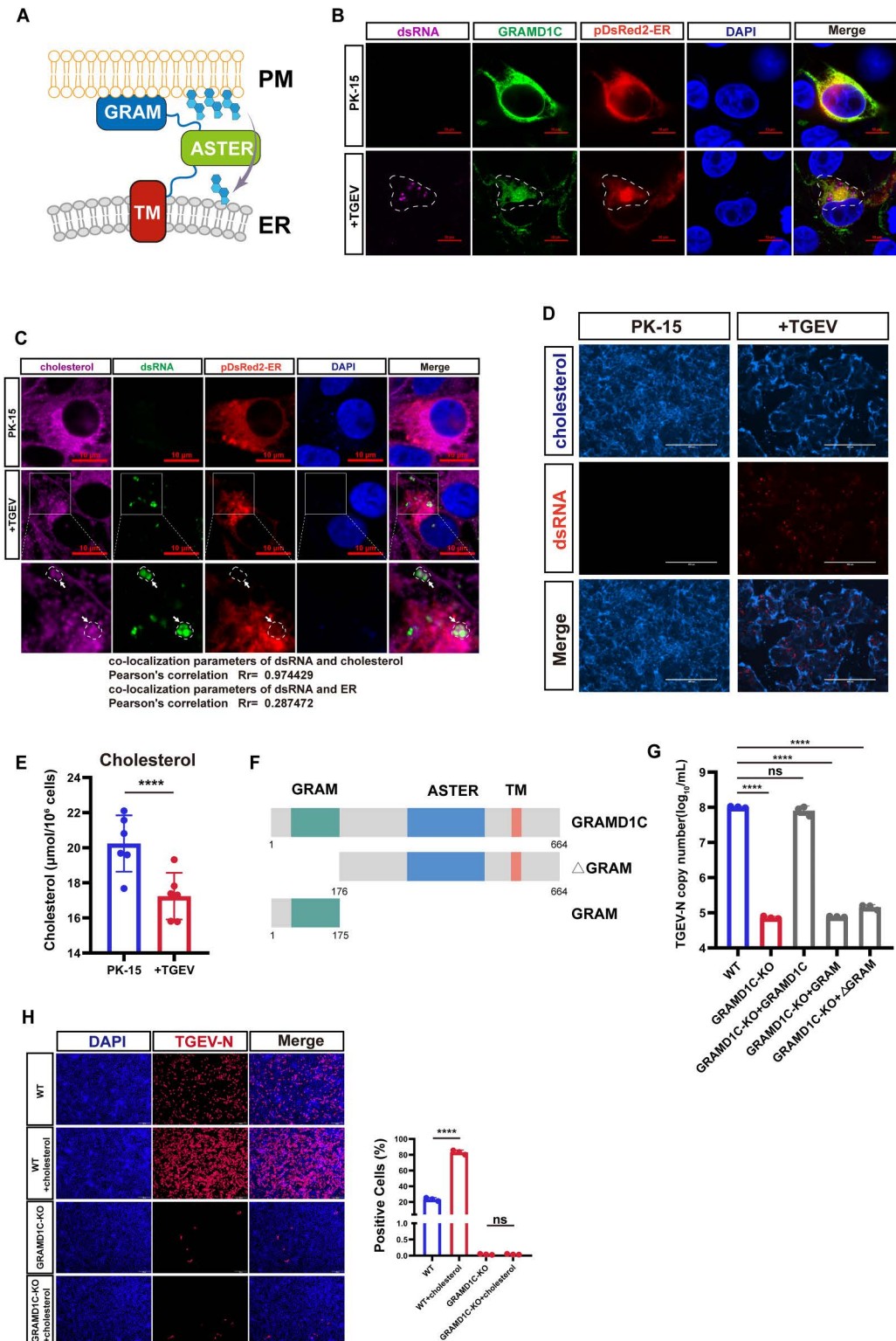

**Fig 4. GRAMD1C-mediated cholesterol transport is critical for viral replication. (A)** Schematic of cholesterol transport by the GRAMD1C ASTER domain from PM to ER. **(B)** Confocal fluorescence microscopy analysis of the co-localization of TGEV dsRNA (indicated in magenta), ectopic expression of GRAMD1C-HA (indicated in green), and ER-tracker pDsRed2 (indicated in red) in WT cells mock-infected or infected with TGEV (MOI = 0.1), at 12 hpi. Scale bars, 10 μm. **(C)** Confocal fluorescence microscopy analysis of the co-localization of free cholesterol (indicated in magenta) and TGEV

dsRNA (indicated in green) in WT cells mock-infected or infected with TGEV (MOI = 0.1), at 12 hpi. Scale bars, 10 μm. The co-localization correlation within the indicated area was analyzed. Pearson's correlation coefficient (Rr) shows a very strong positive correlation between the two channels. **(D)** Filipin III staining of PK-15 cells during TGEV infection (MOI = 0.1) at 24 hpi. Scale bar, 400 μm. **(E)** Free cholesterol levels were determined using a specific assay kit (Solarbio, BC1895) as per the manufacturer's instructions. **(F)** Schematic representation of the construction of full-length GRAMD1C, GRAMD1C lacking the GRAM domain (ΔGRAM), and GRAMD1C containing only the GRAM domain (GRAM). **(G)** RT-qPCR assay for determination of TGEV replication in GRAMD1C-KO cells and cells rescued with different domains, Cells were infected with TGEV (MOI = 0.01) and tested at 24 hpi. **(H)** IFA assay showing TGEV replication (MOI = 0.01) at 18 hpi in WT-PK-15 and GRAMD1C-KO PK-15 cells with or without exogenous cholesterol supplementation. Scale bar, 200 μm. The means and SDs of the results from three independent experiments are shown. ns, not significant; ****$P < 0.0001$. $P$-values were determined by two-tailed unpaired Student's $t$-tests. The data underlying this Figure can be found in S3 Data.

of DMV structures, TEM was performed. HEK-293T cells and PK-15 cells transfected with GRAMD1C, TGEV-nsp3, and TGEV-nsp4 were treated with 10 μM or 30 μM inhibitors, respectively, for 8 h. In both HEK293T and PK-15 cells, 20α-HC treatment significantly reduced DMV numbers (quantified in right panels) (Figs 5G, 5H, and S11C). Notably, inhibitor-treated cells exhibited atypical crescent-shaped membrane structures distinct from canonical DMVs, as indicated by red arrows (Fig 5G). All of these findings establish that the GRAMD1C inhibitor 20α-HC robustly suppresses viral replication by disrupting DMV biogenesis.

To examine the potential broad-spectrum antiviral effect of 20α-HC, we first assessed its impact on HCoV-229E replication. Caco-2 cells were treated with 10 μM 20α-HC following viral infection (S11D Fig). Both IFA and viral titer assays demonstrated a significant suppression of HCoV-229E replication (Fig 5I). Similarly, in Vero-E6 cells treated with 30 μM 20α-HC (S11E Fig) and infected with SARS-CoV-2, a marked inhibition of viral replication was observed (Fig 5J). We further extended our analysis to PEDV, PDCoV, and MHV. Viral titer assays revealed that 20α-HC treatment significantly suppressed the replication of all coronaviruses tested (Figs 5K and S11F). Collectively, these results indicate that 20α-HC acts as a broad-spectrum inhibitor against multiple coronaviruses.

Supported by our data, we propose a model illustrating GRAMD1C's roles in DMV biogenesis (Fig 5L): Viral nonstructural proteins nsp3 and nsp4 localize to ER membranes at DMV formation sites to initiate vesicle generation. In WT cells, when GRAMD1C is present, nsp4 interacts with GRAMD1C, recruiting GRAMD1C to DMV formation sites. GRAMD1C then delivers cholesterol to these sites, enabling the formation of intact DMVs that support viral replication. Conversely, in GRAMD1C-KO or inhibitor-treated cells, viral particles cannot acquire sufficient cholesterol from the ER to form complete DMVs. Instead, only crescent-shaped, incomplete vesicular structures are generated, ultimately inhibiting viral replication.

## GRAMD1C is required for CoV infection in vivo

To assess whether GRAMD1C is crucial for coronavirus replication in vivo, we generated heterozygous GRAMD1C KO mice (GRAMD1C$^{+/-}$) using CRISPR/Cas9 technology, targeting Exons 3 to 4 of the *GRAMD1C* gene (S12A–S12C Fig). Due to the essential role of GRAMD1C, homozygous deletion resulted in embryonic lethality.

We divided GRAMD1C$^{+/-}$ and WT mice (approximately 8 weeks old, $n = 6$ per group) into two groups. Each mouse was intraperitoneally injected with MHV-A59 ($1 \times 10^6$ plaque-forming units, p.f.u.), while control groups received an equivalent volume of DMEM. Weight changes and survival were monitored daily. GRAMD1C$^{+/-}$ mice exhibited a significantly smaller reduction in body weight and delayed disease progression compared to WT mice upon MHV-A59 infection (Fig 6A–6C). Necropsy on the third day post-infection revealed significantly less liver necrosis and tissue damage in GRAMD1C$^{+/-}$ mice compared to WT mice (Fig 6D). Hematoxylin and eosin (H&E) staining of liver tissues showed extensive necrosis and abnormal nuclear aggregation in WT mice, while GRAMD1C$^{+/-}$ mice displayed only focal necrosis (Fig 6E). To assess viral antigen distribution, immunohistochemistry (IHC) with an anti-dsRNA antibody was performed. It revealed stronger positive signals in the liver tissues of WT mice compared to GRAMD1C$^{+/-}$ mice (Fig 6F). Viral titer assays confirmed significantly lower viral loads in the livers of GRAMD1C$^{+/-}$ mice compared to WT mice (Fig 6G).

Given that our previous studies demonstrated the inhibitory effect of 20α-HC on viral replication in vitro, we tested its efficacy in vivo. Mice were administered 20α-HC (20 mg/kg) via intraperitoneal injection 24 hours and 12 hours before, as

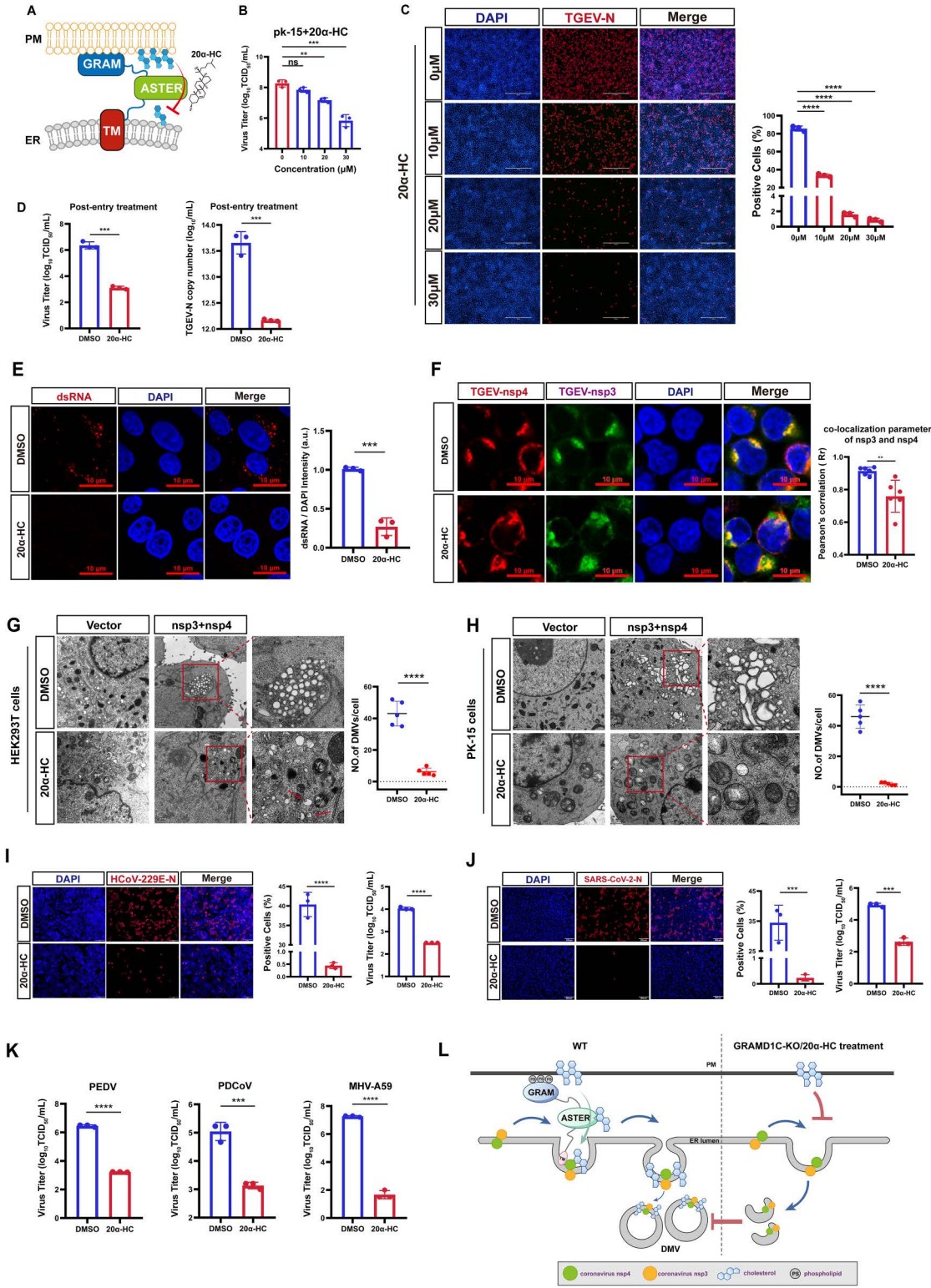

**Fig 5. GRAMD1C inhibitor 20α-HC blocks viral DMV biogenesis. (A)** Schematic of 20α-HC-mediated blockade of cholesterol transport by the GRAMD1C ASTER domain from PM to ER. **(B)** PK-15 cells were incubated with different concentrations of 20α-HC in advance for 8 hours, and then cells were infected with TGEV (MOI = 0.1). Subsequently, the viral titer assay was performed. **(C)** The TGEV N protein was detected by

immunofluorescence assays with different concentrations of 20α-HC. Scale bar, 400 μm. **(D)** PK-15 cells were infected with TGEV (MOI = 0.1) and 20α-HC was added after viral entry for 8 hours. The TGEV titers were tested at 24 hpi. The TGEV N copy number was assessed by RT-qPCR assay at 24 hpi. **(E)** Confocal microscopy to evaluate TGEV replication by detecting dsRNA formation in cells treated with 20α-HC after virus entry. The cells were infected with TGEV (MOI = 1) for 12 hours. Scale bars, 10 μm. **(F)** Co-localization of TGEV-nsp3-HA (green) and TGEV-nsp4-His (red) was analyzed by confocal microscopy in HEK-293T cells with or without inhibitor treatment for 8 hours, after transfection. Scale bar, 10 μm. **(G and H)** TEM analysis demonstrated the presence of DMVs in WT and inhibitor-treated cells following the overexpression of plasmids encoding nsp3 and nsp4. DMVs were manually counted within complete single-cell profiles (n = 5 cell profiles per group). Scale bar, 2 μm. **(G)** HEK293T cells overexpressed with nsp3 and nsp4. Atypical crescent-shaped structures in inhibitor-treated cells are denoted by red arrows. **(H)** PK-15 cells overexpressed with nsp3 and nsp4. **(I)** Caco-2 cells were infected with HCoV-229E (MOI = 0.1), and then cells were incubated with 10μM 20α-HC for 8 hours. The HCoV-229E N protein and virus titers were detected at 36 hpi. **(J)** Vero-E6 cells were infected with SARS-CoV-2 (MOI = 0.01), and then cells were incubated with 30μM 20α-HC for 8 hours. The SARS-CoV-2 N protein and virus titers were detected at 24 hpi. **(K)** Vero-E6 cells were infected with PEDV (MOI = 0.1), and then cells were incubated with 30μM 20α-HC for 8 hours. The PEDV titers were tested at 24 hpi. PK-15 cells were infected with PDCoV (MOI = 5) in the presence of 7.5 μg/ml trypsin, and then cells were incubated with 30μM 20α-HC for 8 hours. The PDCoV titers were tested at 24 hpi. L929 cells were infected with MHV (MOI = 0.01), and then cells were incubated with 30μM 20α-HC for 8 hours, The MHV titers were tested at 18 hpi. **(L)** The proposed model of the roles of GRAMD1C in DMV biogenesis. During the coronavirus replication stage, the viral nsp3 and nsp4 induce zippering and bending of the ER. In this process, nsp4 recruits GRAMD1C to DMV formation sites by binding to its TM domain, utilizing cholesterol delivered to the ER by GRAMD1C for DMV biogenesis. In GRAMD1C-KO cells or inhibitor-treated cells, insufficient cholesterol supply at DMV formation sites results in the generation of only crescent-shaped vesicular structures, thereby inhibiting DMV formation. The means and SDs of the results from three independent experiments are shown. ns, not significant; **$P < 0.01$; ***$P < 0.001$; ****$P < 0.0001$. $P$-values were determined by two-tailed unpaired Student's $t$-tests. The data underlying this Figure can be found in S4 Data.

well as after viral challenge (Fig 6H). Post-infection, 20α-HC-treated mice showed significantly reduced weight loss and delayed onset of morbidity and mortality compared to untreated mice (Fig 6I–6K). H&E staining analysis of liver tissues from 20α-HC-treated mice revealed milder pathological changes and reduced necrosis compared to controls (Fig 6L and 6M). IHC results revealed that, compared with WT mice, the inhibitor-treated mice exhibited a more restricted distribution and weaker intensity of dsRNA signals (Fig 6N). The treated group also exhibited lower viral titers in the liver (Fig 6O).

## Discussion

The emergence of novel coronaviruses continues to pose significant threats to global human and animal health. Clinical research has revealed substantial alterations in lipid metabolism during SARS-CoV-2 infection [36–39]. This suggests a significant link between host lipid metabolism and coronavirus replication. GRAMD1C is known for its crucial role in non-vesicular transport of cholesterol from the PM to the ER, a process essential for maintaining cellular cholesterol homeostasis [40]. Upon reaching a threshold cholesterol level in the PM, the GRAM domain of GRAMD1C interacts with membrane phosphatidylserine, facilitating the establishment of PM-ER contact sites. The ASTER domain then binds and transfers cholesterol from the PM to the ER [32]. While the role of GRAMD1C in cellular cholesterol homeostasis is established, its involvement in coronavirus replication remains unexplored.

Coronaviruses reorganize host membranes to form DMVs, which serve as replication platforms. This process is driven by viral nonstructural proteins (e.g., nsp3, nsp4) that induce membrane curvature and require cholesterol enrichment [41,42]. Previous studies have indicated that cholesterol enrichment within DMVs is crucial for establishing viral replication sites, in both coronaviruses and other viruses such as HCV [23,43]. Our study identifies GRAMD1C, a non-vesicular cholesterol transport protein, as a crucial host factor for multiple coronaviruses, mediating DMV formation. We demonstrate that GRAMD1C co-localizes and interacts with the viral DMV-associated protein nsp3 and nsp4, and that its cholesterol transport activity is essential for viral replication, as evidenced by domain-rescue and inhibitor experiments. These findings position GRAMD1C, through its interaction with nsp3 and nsp4, as a facilitator that establishes ER-DMV contact sites to exploit host cholesterol for viral replication. Notably, the co-localization and interaction patterns of GRAMD1C with nsp3 were similar to those with nsp4, indicating a potential cooperative role for nsp3 and nsp4 in GRAMD1C recruitment. A parallel can be drawn with HCV, which similarly hijacks the VAP protein to establish membrane contacts for replication [44,45]. However, the mechanism by which viral proteins utilize cholesterol after acquiring it from lipid transporters such as

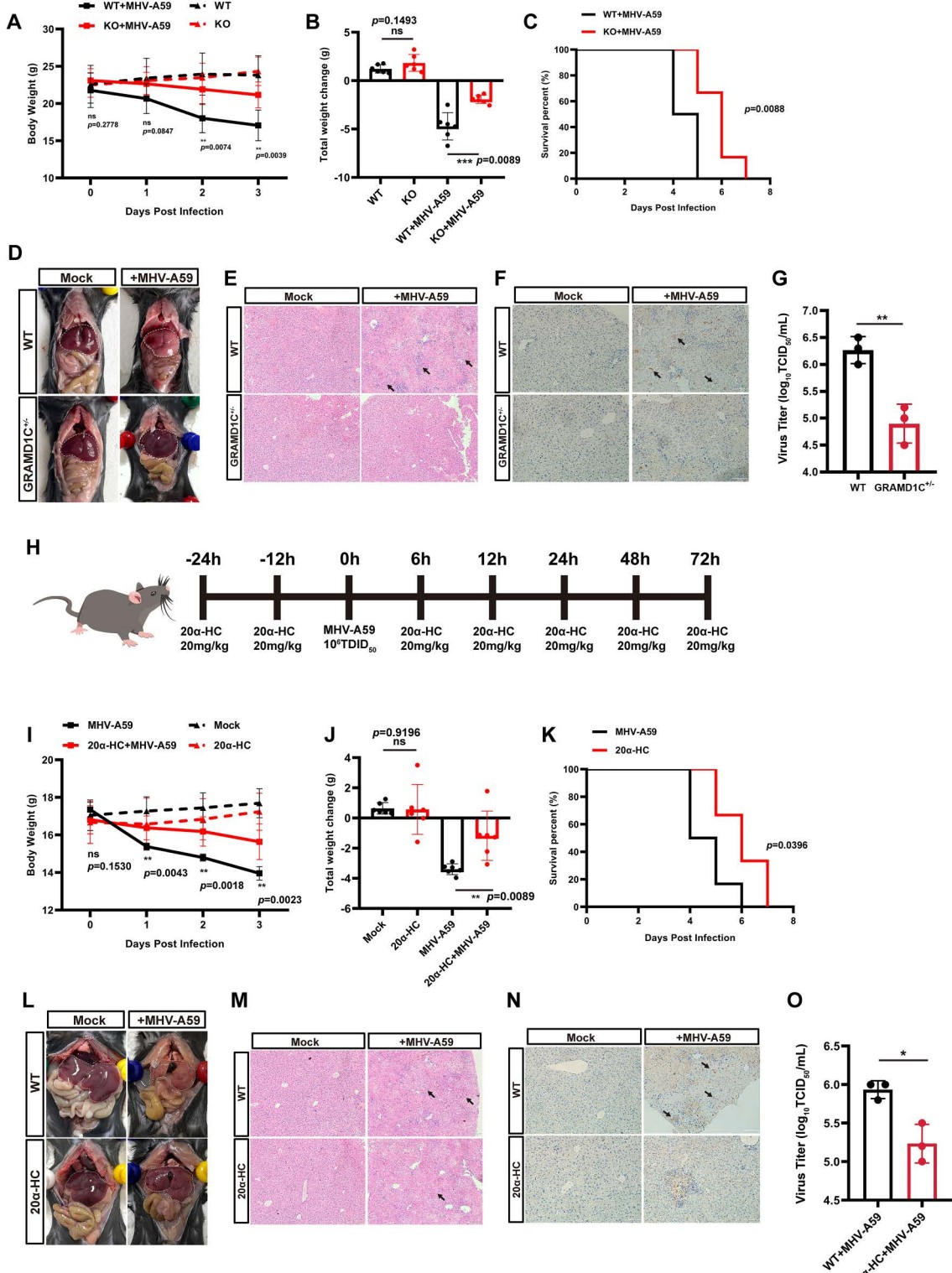

**Fig 6. GRAMD1C is required for CoV infection in vivo. (A)** Line graphs showing the body weight of GRAMD1C[+/−] and WT mice during the 3 days post-inoculation with MHV A59 ($n=6$ mice per genotype). **(B)** The total weight change of GRAMD1C[+/−] and WT mice during the 3 days post-inoculation of MHV A59 ($n=6$ mice per genotype). **(C)** Survival curves for GRAMD1C[+/−] and WT mice infected with MHV A59 ($n=6$ mice per genotype). **(D and E)**

Histopathological analysis of the degree of liver damage in WT and GRAMD1C[+/−] mice infection with MHV A59 at 3 dpi. **(D)** Gross postmortem examination of liver tissue; **(E)** histological examination of Hematoxylin–eosin (H&E) stained liver tissue. The lesion area is indicated by the arrow. Scale bar, 100 μm. **(F)** IHC staining for dsRNA in the liver tissue of WT and GRAMD1C[+/] mice at 3 dpi following infection with MHV A59 (positive signals indicated by arrowheads). Scale bar, 100 μm. **(G)** The virus titers of half livers of GRAMD1C[+/−] and WT mice were measured $TCID_{50}$ assay with L929 cells ($n = 3$ mice per genotype). **(H)** Schematic diagram of the mouse inhibitor experiment. WT Mice were administered 20 mg/kg 20α-HC via intraperitoneal injection 24 hours and 12 hours before, as well as after viral challenge. **(I)** Line graphs showing the body weight of 20α-HC-treated and WT mice during the 3 dpi with MHV A59 ($n = 6$ mice per genotype). **(J)** The total weight change of 20α-HC-treated and WT mice during the 3 days post-inoculation of MHV A59 ($n = 6$ mice per genotype). **(K)** Survival curves for WT and 20α-HC-treated mice infected with MHV A59 ($n = 6$ mice per genotype). **(L and M)** Histopathological analysis of the degree of liver damage in WT and 20α-HC-treated mice infection with MHV A59 at 3 dpi. **(L)** Gross postmortem examination of liver tissue; **(M)** histological examination of Hematoxylin–eosin (H&E) stained liver tissue. Scale bar, 100 μm. **(N)** IHC staining for dsRNA in the liver tissue of WT and 20α-HC-treated mice at 3 dpi following infection with MHV A59 (positive signals indicated by arrowheads). Scale bar, 100 μm. **(O)** The virus titers of half livers of 20α-HC treated and WT mice were measured $TCID_{50}$ assay with L929 cells ($n = 3$ mice per genotype). The means and SDs of the results from three independent experiments are shown. ns, not significant; *$P < 0.05$; **$P < 0.01$; ***$P < 0.001$. $P$-values were determined by two-tailed unpaired Student's $t$-tests. The data underlying this Figure can be found in S5 Data.

GRAMD1C remains unknown. Elucidating this process is critical to directly understanding how cholesterol contributes to the formation of viral DMVs.

The involvement of non-vesicular lipid transport proteins in viral replication has been previously reported. The oxysterol-binding protein (OSBP), which exchanges cholesterol/phosphatidylinositol 4-phosphate (PI4P) at ER-Golgi interfaces, is recruited by SARS-CoV-2 nsp3, nsp4, and nsp6 to supply cholesterol to DMVs [46,47]. While OSBP operates at the ER-Golgi interface, our work reveals GRAMD1C as a distinct host factor operating at PM–ER contact sites, highlighting a spatially complementary cholesterol hijacking pathway, which suggests coronaviruses can flexibly exploit multiple cellular cholesterol sources.

In addition to non-vesicular transporters, NPC1—a vesicular cholesterol transporter that traffics cholesterol from lysosomes to organelles such as the ER—has also been identified as a key host factor for coronavirus replication [29,48–50]. It has been reported that another member of the GRAMD1 family, GRAMD1B, acts at ER-Golgi contact sites to remove excess cholesterol from the Golgi [51]. Therefore, GRAMD1C may integrate cholesterol from both vesicular (NPC1-dependent) and non-vesicular sources to the ER, thereby integrating different transport pathways to facilitate viral replication.

To determine which stage of TGEV proliferation is affected by GRAMD1C, we found that GRAMD1C influences both viral replication and entry. Specifically, loss of GRAMD1C downregulated the expression of *ANPEP* and *ACE2*. For SARS-CoV-2, GRAMD1C appears to play a major role in replication; however, given its effect on *ACE2* expression, a contribution to viral entry cannot be ruled out. In contrast, GRAMD1C did not affect *CEACAM1* expression. We hypothesize that the selective downregulation of certain viral receptors may result from alterations in signaling pathways (e.g., MAPK) induced by GRAMD1C loss.

20α-HC, an inhibitor of GRAMD1C, exhibits structural similarity to the previously reported oxysterol 24HC. Both 24-HC and 20α-HC are cholesterol-derived oxysterols, but their distinct hydroxyl group positions confer different mechanisms and specificities. 24HC primarily disrupts OSBP function, elevating cholesterol levels in multivesicular bodies (MVBs) or late endosomes (LEs) and inhibiting viral entry [52]. In contrast, 20α-HC directly binds to the hydrophobic pocket of the ASTER domain of GRAMD1C, blocking its cholesterol transfer activity and impairing viral DMV formation [35]. This comparison illustrates how different oxysterols can selectively target distinct nodes within the host cholesterol transport network for antiviral effect. Although 20α-HC showed potent cellular activity, its reduced efficacy in vivo highlights the need for further pharmacokinetic optimization to advance its therapeutic potential.

In summary, our study establishes GRAMD1C as a novel and critical host factor for coronavirus replication, operating via a distinct pathway from known transporters like OSBP. We further validate GRAMD1C and its inhibitor 20α-HC as promising targets for developing broad-spectrum antivirals. Future work should focus on elucidating the detailed interactome of GRAMD1C during infection and improving the pharmacological properties of 20α-HC or its analogs.

## Materials and methods

### Ethics statement

All experiments involving mice were performed in accordance with the recommendations in the Guide for the Care and Use of Laboratory Animals of the Ministry of Science and Technology of China and were approved by the Scientific Ethics Committee of Huazhong Agricultural University (permit number: HZAUMO-2024-0234; HZAUMO-2024-0235). Animal care and maintenance protocols complied with the recommendations detailed in the Regulations for the Administration of Affairs Concerning Experimental Animals crafted by the Ministry of Science and Technology of China.

### Plasmid construction

To construct the lentivirus sgRNA expression vector, the lenti-sgRNA-EGFP vector, was digested by the *Bbs*I restriction enzyme (NEB). Individual sgRNAs were cloned into the lenti-sgRNA-EGFP vector for validation in PK-15 cells, Caco-2, Vero cells, and L929 cells, respectively. For rescue and overexpression experiments, to abolish cleavage by sgRNA and Cas9 in GRAMD1C-KO cells, a specific base in the protospacer adjacent motif sequence of GRAMD1C was mutated, which did not alter the encoded amino acid. The mutant sequence was cloned into the pCAGGS-HA vector (Clontech), which was linearized with *EcoR*I and *Kpn*I. For GRAMD1C domain and pApn restoration experiments, the coding sequences of different GRAMD1C domains and pApn were cloned into the pCAGGS-HA vector. Sanger sequencing (Tsingke) was performed to confirm all plasmids. TGEV-BAC, pCAGGS-TGEV-nsp3-HA, pCAGGS-TGEV-nsp4-FLAG plasmids were constructed in our laboratory. All primer sequences are listed in S1 Table.

### Cell culture and viruses

PK-15, Caco-2, and HEK293T cell lines were purchased from the Cell Bank of the Chinese Academy of Sciences (Shanghai, China). Vero-E6 (C1008) and L929 (CCL-1) cell lines were purchased from ATCC (USA). All cell lines were then subjected to mycoplasma detection. For cell culture experiments in the present study, all cells were cultured in Dulbecco's modified Eagle's medium (DMEM) supplemented with 10% fetal bovine serum (FBS), 100 U/mL penicillin, and 100 μg/mL streptomycin and maintained at 37 °C with 5% $CO_2$. The following viruses were used: TGEV WH-1 strain (GenBank accession HQ462571.1), SARS-CoV-2 original strain (IVCAS 6.7512), PDCoV strain CHN-HN-2014 (GenBank accession no. KT336560), DR13(GenBank 282 JQ023162.1), HCoV-229E VR740 (ATCC), and MHV A59 strain (GenBank accession no. MF618253.1). All work involving SARS-CoV-2 has been conducted according to BSL-3 environment safety standards.

### Generation of candidate gene KO cell lines

sgRNA-targeted candidate genes were cloned into the linearized lenti-sgRNA-EGFP, and lentivirus containing sgRNAs were produced as described in a previous study [25]. The sgRNA lentivirus was then transduced into Cas9 cells. On the third day after transduction, cells with GFP expression were enriched by fluorescence-activated cell sorting.

### Transfection and infection

All cell transfections were performed with JetPRIME (PolyPlus) reagent according to the manufacturer's instructions. For viral infections, WT and gene KO PK-15 cells were grown to approximately 80% confluency in a 12-well plate and incubated with TGEV at 37 °C for 1 hour. After adsorption, the cells were washed with phosphate-buffered saline (PBS), and the inoculum was removed and replaced with fresh medium supplemented with 2% FBS, followed by incubation at 37 °C in 5% $CO_2$ for the indicated time. Infection was subsequently monitored by immunofluorescence assays and RT-qPCR assays.

### Western blotting and antibodies

Cells were lysed in cell lysis buffer containing 1× protease inhibitor (Beyotime) for 30 min at 4 °C, and cellular debris was removed via centrifugation (12,000g for 15 min at 4 °C). The lysed cells were denatured with SDS by heating at 95°C

for 10 min, followed by incubation on ice for 5 min. Proteins were separated by SDS-PAGE and transferred onto polyvinylidene fluoride (PVDF) membrane. Membranes were blocked with 5% milk in TBST for at least 3 hours and incubated with primary antibody at 4 °C overnight in 5% milk TBST. Membranes were washed 3× with TBST and incubated with secondary antibodies for 1 hour at room temperature. The following primary antibodies were used: anti-TGEV nucleocapsid protein (rabbit polyclonal antibody, prepared in our laboratory, 1:1,000), anti-GAPDH (AF5009, Beyotime, 1:5,000), anti-GRAMD1C (ab121365, abcam, 1:500), anti-SARS-CoV-2 nucleocapsid protein (#26369, Cell Signaling, 1:1,000), anti-HA (M180-3, MBL, 1:10,000), anti-FLAG (66008-4-Ig, Proteintech, 1:10,000). Secondary antibodies for western blotting used were anti-mouse (no. A21010, Abbkine, 1:10,000) and anti-rabbit (Abbkine, no. A21020, 1:10,000). Proteins were detected with SuperFemto ECL Master Mix detection reagents (Vazyme, China).

## RT-qPCR

Total RNA was extracted from cells with the TRIzol Reagent (Invitrogen). Complementary DNAs (cDNAs) were synthesized using PrimeScript RT Reagent Kit with gDNA Eraser (TaKaRa) in a total volume of 10 μL. Each RT-qPCR reaction was carried out with 100 ng of cDNA and 5 nM primer pairs by using SYBR Green Mix (Bio-Rad, USA). The results were monitored using a CFX96 Real-Time PCR Detection System (Bio-Rad, USA) programmed for one cycle of 15 min at 95 °C, followed by 39 cycles of 10 seconds at 95 °C and 30 s at 60 °C. The relative expression levels were calculated using $2^{-\Delta\Delta Ct}$ method. Beta-actin gene was used as a normalization control. For absolute RT-qPCR, approximately 1 μL of viral RNAs was used as template to synthesize cDNAs. Absolute RT-qPCR assays were performed using SYBR Green Mix and primers specific for the N gene of TGEV in a final reaction volume of 10 μL. The TGEV N protein-coding cDNA sequence from GenBank (accession number: ADY39745.1) was cloned into the pcDNA3.1 vector and used as an internal reference for the quantification of TGEV copy numbers. All primers used in quantitative PCR are listed in S1 Table.

## Viral titers

Candidate gene-KO cells and WT cells were inoculated with virus in 24-well plates in triplicate. After virus infection, the medium was changed after 2 hours of incubation. The cell supernatants were harvested at indicated time point, and viral titers were determined by the median tissue culture infectious dose ($TCID_{50}$) assays using indicated cells.

## Immunofluorescence assay

The protein expression levels of TGEV N protein in candidate gene KO cells and control cells were determined via immunofluorescence assay. Cells were infected with TGEV at different MOIs. At 24 hpi the cells were fixed with 4% paraformaldehyde for 30 min at room temperature and then permeabilized with 0.3% Triton X-100 in PBS for 10 min. Subsequently, the permeabilized cells were blocked for 30 min with PBS containing 5% bovine serum albumin. Cells were incubated with TGEV N antibody (1:1,000), anti-HA antibody (1:1,000) or anti-dsRNA (10010200, Scicons, 1:1,000) for 2 hours and the primary antibodies were recognized by Alexa Fluor 594 Goat anti-Mouse IgG (H + L) (A-11005, Invitrogen, 1:1,000), Alexa Fluor 594 Anti-Rabbit IgG (H + L) (A-11012, Invitrogen, 1:1,000) or Alexa Fluor 488 Anti-Rabbit IgG (H + L) (A-11008, Invitrogen, 1:1,000) at 37 °C for 1 hour. Cell nuclei were counterstained with 4′,6-diamidino-2-phenylindole (DAPI) (D9542, Sigma, 1:1,000) at room temperature for 1 min in the dark. Cell samples were observed and imaged with a fluorescence microscope (Thermo Fisher Scientific EVOS FL Auto). The following antibodies were also used: anti-HCoV-229E nucleocapsid protein (rabbit polyclonal antibody, prepared in our laboratory, 1:1,000), anti-PEDV nucleocapsid protein (rabbit polyclonal antibody, prepared in our laboratory, 1:1,000), anti-PDCoV nucleocapsid protein (rabbit polyclonal antibody, prepared in our laboratory, 1:1,000), CoraLite-647-conjugated HA Tag Recombinant monoclonal antibody (CL647-81290, Proteintech, 1:100), FITC-conjugated DYKDDDDK tag Monoclonal antibody (FITC-66008, Proteintech, 1:100), CoraLite-594-conjugated 6*His-Tag Monoclonal antibody (CL594-66005, Proteintech, 1:100).

The mean fluorescence intensity was measured using ImageJ (National Institutes of Health, Bethesda, MD, USA) for both the DAPI and dsRNA channels within defined regions of interest. The ratio of the dsRNA channel intensity to the DAPI channel intensity was then calculated and used for comparative analysis.

Quantitative analysis of the percentage of positive cells was performed using ImageJ software. For each region of interest (ROI), images were converted to 8-bit grayscale. A consistent threshold was manually set to distinguish positive signal from background, and this threshold was applied uniformly to all images within the same experimental set. The "Analyze Particles" function was then used to count positively stained cells. The total number of cells in the same ROI was determined by DAPI. The percentage of positive cells was calculated as (Number of positive cells/ Total number of cells) × 100%. For each condition, at least three independent biological replicates were analyzed, with multiple ROIs quantified per replicate. Data are presented as mean ± standard deviation (SD). Statistical significance was determined using two-tailed unpaired Student's $t$-tests, with a $p$-value < 0.05 considered significant.

## MTS and EdU cell proliferation assay

To assess cell proliferation, MTS and EdU cell proliferation assays were performed. The MTS assay was performed using MTS Assay Kit (ab197010, Abcam). Cells were seeded into 96-well plates and allowed to adhere for 12 hours. For inhibitor-treated groups, cells were incubated with the indicated concentrations of inhibitor for 8 hours. Subsequently, 10 μL of MTS reagent was added directly to each well (containing either KO cells or pre-treated cells) and incubated at 37 °C for 4 h in standard culture conditions. The plate was briefly shaken, and the absorbance of treated and untreated cells was measured using a plate reader at an optical density (OD) of 490 nm.

For the EdU assay, cells were seeded into 24-well plates. After 24 h of incubation, the EdU cell proliferation assay was performed using BeyoClick EdU Cell Proliferation Kit (C0075S, Beyotime) with Alexa Fluor 555 according to the instruction. The cell nuclei were stained with DAPI (C1005, Beyotime) at room temperature for 1 min in the dark. Stained cells were visualized under a fluorescence microscope. The proportion of EdU-positive cells was calculated using Image J software (three independent wells were imaged, and one random field of view per well was captured for each experimental phase). The proportion of EdU-positive cells was quantified using the same method as described for the immunofluorescence assay.

## TEM assay

Cells were infected with TGEV at an MOI of 1 for 12 hours. For the transfected cells, cells were infected with virus at 24 hours post-transfection. The cells were washed three times with precooled PBS and fixed by adding 3 mL of 2.5% glutaraldehyde (Servicebio) at room temperature for 2 h. After fixation, cells were transferred to a 2 mL centrifuge tube. TEM samples were performed by Servicebio Company, and images were taken using an HZAU TEM platform (no. H7650, HITACHI).

TEM images were quantified for DMV abundance. For each condition, multiple random cell profiles were analyzed. Within the boundary of each entire single-cell profile, all DMVs were manually counted.

## Virion attachment and internalization assay

WT and GRAMD1C-KO cells were infected with TGEV (MOI = 50) and incubated at 4 °C for 1 hour. For the virus attachment assay, cells were washed three times with cold PBS (at 4 °C) to remove unbound viral particles and harvested to extract viral RNA. The amount of viral RNA was determined by RT-PCR. For the internalization assay, the infected cells described above were further cultured with prewarmed DMEM at 37 °C for another 30 min. Subsequently, the cells were treated with 1 mg/mL pronase in cold PBS to remove the attached but uninternalized viral particles. After washing three times, the cells were lysed with TRIzol reagent to extract total RNA, and the viral RNA was quantified by RT-PCR.

## Confocal microscopy

To observe the early viral replication of the TGEV in host cells, the same amount of GRAMD1C-KO+ANPEP cells and control cells were cultured in 35-mm Petri dishes overnight. The equivalent dose of TGEV (MOI = 50) was added to each well and incubated at 4 °C for 60 min for complete adsorption and then transferred to 37 °C and incubated for 3 hours. Immunofluorescence assays were performed as described above, while the images were acquired using a laser scanning confocal microscope (Nikon). The subcellular localization of GRAMD1C in WT cells transfected with pCAGGS-GRAMD1C-HA plasmid was observed after 24 hours. Afterward, TGEV (MOI = 0.1) was added into these cells, incubated for 12 hours, immunolabeled with a HA-tag antibody (M180-3, MBL, 1:1,000) and dsRNA antibody (SCICONS, no. 10010200, 1:1,000), and imaged to identify double fluorescent positive cells.

The subcellular localization of TGEV nsp3, TGEV nsp4 and GRAMD1C in WT cells transfected with pCAGGS-nsp3-HA, pCAGGS-nsp4-Hia, pCAGGS-GRAMD1C-FLAG plasmids was observed after 24 hours, immunolabeled with 647-conjugated HA tag recombinant antibody (CL647-81290, proteintech,1:500), FITC-conjugated FLAG tag monoclonal antibody (FITC-66008, proteintech,1:500) and 594-conjugated His-tag monoclonal antibody (CL594-66005, proteintech,1:500).

## Protein–protein docking

The proteins used for docking were porcine GRAMD1C(A0A8D1XGN7), human GRAMD1C(Q8IYS0), TGEV-nsp4(P0C6V2), SARS-CoV-2-nsp4 (P0C6X7). The HDOCK SERVER (http://hdock.phys.hust.edu.cn/) was employed for protein-protein molecular docking [53]. The binding sites of the proteins are presented in S2 Table-1. The MM/GBSA mode of the HawkDOCK SERVER (http://cadd.zju.edu.cn/hawkdock/) was utilized to calculate the free binding energy of protein-protein interactions and rank the free binding energy of amino acid residue sites (S2 Table-2) [54]. The docking score was adopted as the evaluation criterion for docking. Among the 10 conformations output by docking, the best docking model was selected and the binding energy was calculated. Finally, we used Pymol 2.4 software to visualize the binding sites with high binding energy and mutual binding. In this way, we could visually observe the binding situation between the proteins.

## Drug treatment assay

20(S)-Hydroxycholesterol (MCE HY-12316) was dissolved in dimethyl sulfoxide at a stock concentration of 10 mM and then frozen in aliquots at −80 °C. Cells were incubated with the specified concentration of inhibitor after infected with virus.

## Free cholesterol staining and quantification

For Filipin staining of free cholesterol, fixed cells were stained with Cholesterol Assay Kit (Cell-Based) (Abcam, ab133116) at room temperature for 2 hours. For antibody staining with Filipin, cells were not permeabilized since Filipin permeabilizes cells. Primary and secondary antibody staining was completed after Filipin staining, but in the absence of saponin. For CY5-cholesterol staining of free cholesterol, cells were stained with 5 μM CY5-cholesterol (Xi'an QIYUE BIOLODY) for 3h before fixation.

Intracellular free cholesterol levels were measured using a Free Cholesterol Content Assay Kit (Solarbio, BC1895) following the manufacturer's instructions.

## Cholesterol-loaded assay

Cells were cultured and transfected in DMEM supplemented with 5% lipoprotein deficient serum (LPDS) (CelliGent). At 24 h post-transfection, the medium was switched to DMEM containing 5% LPDS with or without 200 μM cholesterol/cyclodextrin (Sigma, C4951). Cells were imaged 65 min following cholesterol addition.

## Generation of GRAMD1C-KO mice and MHV challenge

The GRAMD1C-KO mice in the C57BL/6 genetic background were generated by Cyagen (China). To generate KO mice, a gDNA fragment spanning Exons 3–4 of GRAMD1C was deleted by co-injecting sgRNAs and Cas9 mRNA into fertilized mouse eggs. The founders were genotyped by PCR amplification of gDNA extracted from mouse tails. WT and GRAMD1C-KO mice (approximately 8 weeks old, $n=6$) were subjected to intraperitoneal injection with MHV-A59 (p.f.u. = $5 \times 10^5$). For the in vivo inhibitor mice experiment, mice were pre-treated with 20 mg/kg 20α-HC before MHV challenge. The control group received an equal volume of DMEM via the same strategy. The weight of each mouse was monitored daily, and the lethality was recorded. The mock and virus-infected mice were sacrificed at the indicated days after infection to observe the liver's pathology. Liver sections were fixed in buffered formalin (4%) solution, embedded in paraffin, and sectioned into 4 μm tissue sections. Sections were stained with H&E for histology, or with an antibody against dsRNA for IHC, then examined and imaged under a light microscope. Livers were homogenized in 1 mL of DMEM and centrifuged at 8,000 rpm for 10 min at 4 °C. The virus titers in the supernatants were collected and measured via $TCID_{50}$ assays with L929 cells.

## Statistical analysis

Statistical significance values were assessed using GraphPad Prism 8.0. Two-tailed unpaired t-tests were used to compare two unpaired groups. Multiple groups were compared by two-way analysis of variance. Unless otherwise stated, the data represent the mean ± SD of experiments performed, at least, in triplicate.

## Supporting information

**S1 Fig. Genomic sequencing and cell viability validation of various GRAMD1C knockout cells. (A)** GRAMD1C-KO PK-15 and WT cells were seeded into 96-well plates, and cell proliferation and viability were validated by MTS assay. **(B)** Proportion of EdU-positive cells between the GRAMD1C-KO PK-15 cells and WT cells. Left: representative pictures. Right: quantification of EdU-positive cells. **(C)** Genomic sequence analysis of *GRAMD1C* in GRAMD1C-KO Vero-E6 cells. **(D)** Cell viability of GRAMD1C-KO Vero-E6 cells was validated by MTS assay. **(E)** Genomic sequence analysis of *GRAMD1C* in GRAMD1C-KO Caco-2 cells. **(F)** Cell viability of GRAMD1C-KO Caco-2 cells was validated by MTS assay. **(G)** Genomic sequence analysis of *GRAMD1C* in GRAMD1C-KO L929 cells. **(H)** Cell viability of GRAMD1C-KO L929 cells was validated by MTS assay. The means and SDs of the results from three independent experiments are shown. ns, not significant. The data underlying this Figure can be found in S5 Data.
(TIF)

**S2 Fig. Analysis of receptor expression levels in GRAMD1C-KO cell lines. (A)** western blot analysis validated the protein expression level of endogenous ANPEP in WT, KO, and GRAMD1C-rescued cells. **(B)** Relative *ANPEP* mRNA levels in WT and GRAMD1C-KO Caco-2 cells, normalized to *β-actin*. **(C)** Relative *ACE2* mRNA levels in WT and GRAMD1C-KO Vero-E6 cells, normalized to *β-actin*. **(D)** Relative *CEACAM1* mRNA levels in WT and GRAMD1C-KO L929 cells, normalized to *β-actin*. The data underlying this Figure can be found in S5 Data and S1 Raw Images.
(TIF)

**S3 Fig. Genomic sequencing and cell viability validation of ANPEP-KO and ANPEP-GRAMD1C-KO cell lines. (A)** Genomic sequence analysis of *ANPEP* in ANPEP-KO and ANPEP-GRAMD1C-KO cells. **(B)** Cell viability of ANPEP-KO and ANPEP-GRAMD1C-KO cells assessed by MTS assay. The data underlying this Figure can be found in S5 Data.
(TIF)

**S4 Fig. Transfection efficiency validation of ANPEP-KO and ANPEP-GRAMD1C-KO cell lines. (A)** The indicated KO cell lines were transfected with a pLVX-mCherry control plasmid. Left: representative images. Scale bar, 400μm. Right: quantification of mCherry-positive) cells. **(B)** Cells were seeded at equal density and transfected with the pRL-TK plasmid.

Luciferase activity, measured at 465 nm, was determined in ANPEP-KO and ANPEP-GRAMD1C-DKO cells to control for transfection efficiency. The data underlying this Figure can be found in S5 Data.
(TIF)

**S5 Fig. Analysis of the viral entry stage in GRAMD1C-KO+ΔNsPEP cell lines. (A)** WT and GRAMD1C-KO cells were infected with TGEV (MOI = 50) at 4 °C for 1 h and assessed for TGEV adsorption. The cells were harvested, and viral RNA was extracted to determine virion attachment at the cell surface. **(B)** WT and GRAMD1C-KO cells were infected with TGEV (MOI = 50) at 4 °C for 1 hour, followed by incubation at 37 °C for 30 min. The internalization was evaluated by absolute RT-qPCR. The data underlying this Figure can be found in S5 Data.
(TIF)

**S6 Fig. Subcellular localization of GRAMD1C in DMVs after cholesterol loaded. (A)** Quantification of intracellular free cholesterol after cholesterol loaded. **(B)** Cells transfected with GRAMD1C and TGEV-nsp3/4 plasmids were switched to 5%LPDS medium ± 200 μM cholesterol for 1 hour prior to fixation and staining. The data underlying this Figure can be found in S5 Data.
(TIF)

**S7 Fig. Co-IP validation of the interaction between GRAMD1C and nsp3/nsp4. (A)** Co-IP analysis of GRAMD1C point mutants (F117A, I559A, L566A) with TGEV-nsp4. **(B)** Co-IP analysis of GRAMD1C point mutants (L595A, L642A, F649A) with TGEV-nsp4. **(C)** Co-IP assay was performed in HEK293T cells transfected with pCAGGS-GRAMD1C-FLAG and pCAGGS-TGEV-nsp3-HA for 24 hours. The data underlying this Figure can be found in S1 Raw Images.
(TIF)

**S8 Fig. Western blot analysis of GRAMD1C and ΔTM-GRAMD1C expression levels.** Western blot confirmed the successful rescue of full-length GRAMD1C and △TM-GRAMD1C in GRAMD1C-KO cells. GAPDH used as an internal control gene. The data underlying this Figure can be found in S1 Raw Images.
(TIF)

**S9 Fig. Confocal fluorescence microscopy analysis of the co-localization of free cholesterol and TGEV dsRNA in WT cells.** Two additional exemplary cells of Fig 4C. Confocal fluorescence microscopy analysis of the co-localization of free cholesterol and TGEV dsRNA in WT cells mock-infected or infected with TGEV (MOI = 0.1), at 12 hpi. Scale bars, 10 μm.
(TIF)

**S10 Fig. Western blot analysis of GRAMD1C and different domains expression levels.** Western blot confirmed the successful rescue of full-length GRAMD1C and different domains in GRAMD1C-KO cells. GAPDH is used as an internal control gene. The data underlying this Figure can be found in S1 Raw Images.
(TIF)

**S11 Fig. Cell viability validation in different cell lines treated with varying concentrations of 20α-HC. (A)** PK-15 cells were treated with the indicated concentrations of 20α-HC for different time points prior to MTS analysis. **(B)** RT-qPCR assay for determination of relative mRNA levels of *ANPEP* in WT and 20α-HC-treated cells at indicated time points. **(C)** HEK-293T cells were treated with the indicated concentrations of 20α-HC for different time points prior to MTS analysis. **(D)** Caco-2 cells were treated with the indicated concentrations of 20α-HC for different time points prior to MTS analysis. **(E)** Vero-E6 cells were treated with the indicated concentrations of 20α-HC for different time points prior to MTS analysis. **(F)** L929 cells were treated with the indicated concentrations of 20α-HC for different time points prior to MTS analysis. The data underlying this Figure can be found in S5 Data.
(TIF)

**S12 Fig. Construction and validation of GRAMD1C heterozygous knockout mice. (A)** Targeting Strategy of two sgRNAs of GRAMD1C-deficient mice. **(B)** A schematic diagram of generation of F2 GRAMD1C -deficient mice. **(C)** The results of tail piece (2–5 mm) PCR detection were consistent with the expected genetic profile of the mouse strain. Heterozygotes: two bands with 736 bp and 692 bp, Wild-type allele: one band with 692 bp.
(TIF)

**S1 Table. Primer pairs and sgRNA targeting sequences used in this study.**
(XLSX)

**S2 Table. Protein–protein docking data between GRAMD1C and nsp4.**
(XLSX)

**S1 Data. Raw data for** Fig 1**.** This compressed folder contains the underlying numerical data and/or uncropped images used to generate the panels in Fig 1.
(ZIP)

**S2 Data. Raw data for** Fig 2**.** This compressed folder contains the underlying numerical data and/or uncropped images used to generate the panels in Fig 2.
(ZIP)

**S3 Data. Raw data for** Figs 3I **and** 4**.** This compressed folder contains the underlying numerical data and/or uncropped images used to generate the panels in Figs 3I and 4.
(ZIP)

**S4 Data. Raw data for** Fig 5**.** This compressed folder contains the underlying numerical data and/or uncropped images used to generate the panels in Fig 5.
(ZIP)

**S5 Data. Raw data for** Figs 6 **and** S1–S6 **and** S11**.** This compressed folder contains the underlying numerical data and/or uncropped images used to generate the panels in Figs 6 and S1–S6, and S11.
(ZIP)

**S1 Raw Images. Original images for all blots and gels in the main and supplementary figures.** This PDF file contains the original scans of all blots and gels presented in the main figures and supplementary information. Relevant bands or areas are clearly marked where applicable.
(PDF)

## Acknowledgments

We thank Yingying Xu (Huazhong Agricultural University) for TEM support. Data from Electron Microscope was acquired at the public instrument center of the College of Animal Science and Technology and College of Veterinary Medicine at Huazhong Agricultural University. We thank Yan Wang (Institute of Hydrobiology, Chinese Academy of Sciences) for flow cytometry technical support. We thank Yulan Zhang for her contribution to preliminary work.

## Author contributions

**Conceptualization:** Zhelin Su, Limeng Sun, Guiqing Peng.

**Data curation:** Zhelin Su, Shengsong Xie, Limeng Sun, Guiqing Peng.

**Formal analysis:** Zhelin Su, Zhen Fu, Yichen Yang, Yanan Fu, Jingwen Mo, Juan Xu, Yubei Tan, Yuejun Shi, Shengsong Xie, Limeng Sun, Guiqing Peng.

**Funding acquisition:** Limeng Sun, Guiqing Peng.

**Methodology:** Zhelin Su, Yubei Tan, Shengsong Xie, Limeng Sun.

**Project administration:** Zhelin Su, Limeng Sun, Guiqing Peng.

**Resources:** Limeng Sun, Guiqing Peng.

**Supervision:** Yuejun Shi, Limeng Sun, Guiqing Peng.

**Validation:** Zhelin Su, Zhen Fu, Yichen Yang, Yanan Fu, Jingwen Mo, Juan Xu, Yixin Xiang.

**Visualization:** Zhelin Su, Limeng Sun.

**Writing – original draft:** Zhelin Su, Limeng Sun.

**Writing – review & editing:** Zhelin Su, Limeng Sun, Guiqing Peng.

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
