## [Editor Report · Decision Letter 0]

7 Aug 2025

Dear Dr Guiqing,

Thank you for submitting your manuscript entitled "Non-vesicular cholesterol transporter GRAMD1C as a pan-coronavirus antiviral target" for consideration as a Research Article by PLOS Biology.

Your manuscript has now been evaluated by the PLOS Biology editorial staff, as well as by an academic editor with relevant expertise, and I am writing to let you know that we would like to send your submission out for external peer review.

Once your full submission is complete, your paper will undergo a series of checks in preparation for peer review. After your manuscript has passed the checks it will be sent out for review. To provide the metadata for your submission, please Login to Editorial Manager (https://www.editorialmanager.com/pbiology) within two working days, i.e. by Aug 09 2025 11:59PM.

Kind regards,

Melissa

Melissa Vazquez Hernandez, Ph.D.

Associate Editor

PLOS Biology

---

## [Decision Letter · Decision Letter 1]

2 Sep 2025

Dear Dr Guiqing,

Thank you for your patience while your manuscript "Non-vesicular cholesterol transporter GRAMD1C as a pan-coronavirus antiviral target" was peer-reviewed at PLOS Biology. It has now been evaluated by the PLOS Biology editors, an Academic Editor with relevant expertise, and by two independent reviewers.

In light of the reviews, which you will find at the end of this email, we would like to invite you to revise the work to thoroughly address the reviewers' reports. As you will see below, majority of reviewers are positive about the relevance and novelty of the study, yet some concerns have raised during revision. Reviewer 1 thinkts that while GRAMD1C clearly affects animal coronaviruses, its impact on SARS-CoV-2 is modest, and reduced ANPEP expression suggests that entry inhibition may be the primary mechanism. The reviewer recommends clarifying this point by testing additional human coronaviruses and lung cell models, and addressing discrepancies between knockout and inhibitor effects. Reviewer 2 thinks there are far-reached statements and that the study has some technical flaws. As R1, s/he also mentions the modest effect on SARS-CoV-2 and recommends testing 20α-HC against SARS-CoV-2 in lung cells, clarifying whether effects on porcine viruses stem from reduced ANPEP receptor expression rather than DMV disruption, and explaining discrepancies between single vs double knockouts and inhibitor results. This reviewer also requests the inclusion of proper controls, quantification of imaging data, and clearer figure labeling, and mentions that additional experiments, such as ELISPOT assays, cholesterol binding/affinity studies, rescue with exogenous cholesterol, or immunofluorescent/TEM analyses, would strengthen mechanistic claims. We agree with all reviewer concerns and would require some additional experimental revisions to address them, as we consider that this would strengthen the work.

Given the extent of revision needed, we cannot make a decision about publication until we have seen the revised manuscript and your response to the reviewers' comments. Your revised manuscript is likely to be sent for further evaluation by all or a subset of the reviewers.

**IMPORTANT - SUBMITTING YOUR REVISION**

*Re-submission Checklist*

*Published Peer Review*

*PLOS Data Policy*

*Blot and Gel Data Policy*

Sincerely,

Melissa

Melissa Vazquez Hernandez, Ph.D.

Associate Editor

PLOS Biology

REVIEWERS' COMMENTS

Reviewer #1:

Su and colleagues provide evidence that the cholesterol transport protein GRAMD1C is essential for coronavirus replication, including SARS-CoV-2. They used animal coronaviruses as models to show that GRAMD1C facilitates double-membrane vesicle formation through interaction with nsp4 and propose that GRAMD1C is a suitable targetable host protein that's required for the development of antivirals. This paper is informative and will be of interest to the wider coronavirus research community.

Major:

-Although GRAMD1C depletion markedly reduces TGEV, PEDV, and PDCoV replication, its impact on SARS-CoV-2 is minimal (from ~6 × 10¹² to ~2 × 10¹² copies/mL). How does this translate into changes in viral titers? What is the effect on endemic human coronaviruses such as HCoV-229E, particularly in light of the pronounced reduction of ANPEP? Considering the distinct tropisms of human versus animal coronaviruses, it would also be valuable to assess these effects in human lung cell lines to better reflect physiological relevance.

-GRAMD1C KO markedly reduced ANPEP expression, the receptor for TGEV and PDCoV, suggesting that disruption of viral entry contributes to the observed decrease in viral titers. Consistently, restoring ANPEP expression in GRAMD1C KO cells diminished the antiviral effect (i.e., inhibition was significantly weaker upon ANPEP reconstitution). This strongly supports a central role of entry inhibition in the antiviral function of GRAMD1C. By contrast, the authors claim that GRAMD1C impacts intracellular viral replication independently of viral entry. However, in the double KO (ANPEP and GRAMD1C), only a minor reduction in viral replication was observed compared with the single ANPEP KO, further underscoring the importance of entry inhibition as a key pathway. While there is an effect on DMV formation, as demonstrated by interaction and Co-IP studies this reviewer is doubtful that this is the major antiviral mechanism. The authors might want to carefully rephrase their conclusions, considering viral entry as additional major mechanism.

-Inhibitor treatment (20�-HC) strongly reduced MHV titers (much more compared to GRAMD1C KO) while the effect on TGEV stayed in a similar range to the previous observations. How do the authors explain these discrepancies? Could there be additional targets/mechanisms? What is the effect of the inhibitor on SARS-CoV-2 using lung cell culture models?

Minor:

-The sentence in the discussion: "Our study provides compelling evidence that GRAMD1C, a non-vesicular cholesterol transport protein, is a critical host factor for the replication of multiple coronaviruses, including SARS-CoV-2, TGEV, PEDV, PDCoV, and MHV-A59.", is an overstatement given the discussion points above and needs rephrasing, specifically given the limitations (strong entry effects, minor effect on SARS-CoV-2 replication).

-Intro: nsp3 and nsp4 are indeed central, but evidence suggests nsp6 also contributes to membrane remodeling, so calling nsp3+nsp4 the minimal viral factors may oversimplify.

-How do the authors explain the faint GRAMD1C band in the KO condition in Fig. 1B? In other words: how stable/complete is the KO? Could these be residual wild-type cells?

-Cell viability should be tested at multiple timepoints post-treatment (ideally up to 48 h).

-How were numbers of DMVs per cell determined? Manually or computationally? How many cells were assessed? The experimental details should be provided. Same for the determination of the normalized mean intensity of IF images (see Figs. 2, 4, 5) and the quantification of EdU-pos. cells.

-What program was used for the protein-protein interaction modeling (Fig. 3C and G)? Include in M&Ms or Fig. legend.

-Mock (WT) controls is missing in the data presented in Fig. 3I, Fig. 4F.

-Fig. 6: histopathology and immunohistochemistry to assess viral antigen distribution would be an informative add-on!

-It would be great to confirm the mouse genotypes by PCR if the authors aimed for clearly separable band sizes (See Fig. S6) - or run a high-resolution agarose gel?

-The whole manuscript should be proofread for grammar and typos.

-Check all figure labels again for consistency in capitalization.

Reviewer #2:

In this study, Su et al. report the non-vesicular cholesterol transporter GRAM domain containing 1C (GRAMD1C) from the Aster protein family as an important host factor supporting the replication of different coronaviruses, i.e. porcine transmissible gastroenteritis virus (TGEV), porcine delta coronavirus (PDCoV), porcine epidemic diarrhea virus (PEDV), murine hepatitis virus (MHV) and, to a much lesser extent, severe acute respiratory syndrome virus 2 (SARS-CoV-2). These viruses share a similar strategy of replicating in ER-derived double membrane vesicles (DMVs). The authors provide compelling evidence that GRAMD1C depletion or pharmacological inhibition impairs viral RNA replication and appears to affect DMV production. Additionally, the authors validated the potent GRAMD1C inhibitor 20α-hydroxycholesterol (20α-HC) as a potential inhibitor impairing the replication of coronaviruses, at least porcine ones and MHV, in cell culture and the latter also in a mouse model. Moreover, a heterozygous deletion of the GRAMD1C gene in mice attenuated MHV replication and reduced pathogenicity in these animals.

Overall, the data are strong and they reveal an important role of the cholesterol transporter GRAMD1C in coronavirus replication, possibly by contributing to the formation of DMVs. A strength of this study is the use of different viruses and cell models and the demonstration of antiviral activity in vivo either by depletion of GRAMD1C expression in animals or inhibition of the enzyme. Nevertheless, I found a couple of instances where far-reaching statements have been made that are not supported by the data as well as a few technical flaws that should be fixed.

Major points:

* The authors frequently switch between different viruses, most of them being porcine viruses. While the phenotype exerted by GRAMD1C interference is very strong on these viruses, and for MHV (around 100-fold reduction of viral replication), this is not the case with SARS-CoV-2 where the effect is just 3-fold (Fig. 1H). This is a major difference raising questions about physiological relevance for SARS-CoV-2. Such further evidence could be obtained by including a test of 20α-HC against SARS-CoV-2 that unfortunately is missing in this study, but needs to be done.

* In Figure 1, the authors make claims that GRAMD1C depletion (KO) affects viral replication. However, in Figure S2A they show that this KO virtually depleted the viral receptor ANPEP from the cells. If receptor expression is so much reduced by GRAMD1C KO, how can the authors exclude that the phenotypes see in Fig. 1 are not caused by reduced receptor expression? Notably, the effect of KO on SARS-CoV-2 is very small whereas the effect on the porcine viruses is very strong. Therefore, the expression of ANPEP needs to be confirmed in all used porcine cell lines given the role of ANPEP in the entry of at least TGEV and PDCoV (https://doi.org/10.1016/j.virol.2019.12.007) and the same needs to be done for the MHV and SARS-CoV-2 receptors. Although the subsequent data show that RNA replication appears to be the main target of GRAMD1C, I find it disturbing to first show the data in Fig. 1 and then revealing that the KO has such a strong effect on receptor expression and thus, (also) virus entry.

* Figure 2B: Why is there a further reduction in the double KO cells? The difference between ANPEP-KO and the double-KO is very moderate (2-3 fold) when comparing with the phenotypes reported in Figure 1 (100-fold) although with the BAC transfection, entry should have been bypassed. So why does depletion of the receptor impair so much viral replication and why is the difference between the single- and double-KO so minor? That needs to be explained.

* Moreover, a transfection control is missing in Figure 2 to exclude that the double-KO cells were indeed transfected. Otherwise, one cannot exclude that the reduction is due to reduced transfection efficiency, rather than the KO itself.

* Figure 2D: I find this data set not very meaningful, because replication is also very much reduced; and reduced replication also means less viral proteins and therefore, also less DMV. Only the data in Fig. 5 allow to draw the authors' conclusion. Either the authors add a statement to that or I suggest to remove this data from Figure 2.

* From the data in Figure 3, the authors conclude that "Collectively, these results demonstrate that GRAMD1C engages coronavirus nsp4 via its TM domain to promote DMV biogenesis……". This is not formally shown, the only point that can be made is that membrane association of GRAMD1C is required to support viral replication. The authors cannot exclude that the TM is necessary to recruit the protein to the membrane, but the interaction occurs via some other regions in the protein. Therefore, one can only state that the TM domain is required for the interaction.

* How is the cholesterol recruitment timed with respect to DMV formation? The authors show an interaction of GRAMD1C with nsp4, which indicates a recruitment of GRAMD1C (and thereby cholesterol) to the DMVs. In the assembled DMVs, nsp4 is located mainly on the inner bilayer of the DMV, so how would GRAMD1C be able to exert its cholesterol-recruitment function? An interesting experiment would be analogous to Sandhu et al. (Figure 3/S4, DOI: 10.1016/j.cell.2018.08.033) to see whether cholesterol-dependent relocalization of GRAMD1C / Aster to the plasma membrane can still happen under conditions of DMV formation, or whether the nsp-associated GRAMD1C portion is "trapped" on the inner DMV bilayer.

* Figure 4C: The authors show intense cholesterol signal in dsRNA punctae and claim specific cholesterol recruitment to replication sites. Since DMV formation goes hand in hand with significant membrane remodelling, this experiment does not indisputably support the authors' statement. Specificity of cholesterol recruitment rather than cholesterol accumulation as a result of membrane remodeling should be demonstrated either by staining an ER-resident lipid species that does not show this enrichment or by using nsp3/4 expressing cells, where upon GRAMD1C-KO the cholesterol enrichment in nsp4-stained punctae should be abrogated.

Minor points:

* In the introduction, the authors frequently switch between different viruses, but make claims that read like being universally applicable to coronaviruses. For instance, the authors state: "As a key lipid component of ER membranes, cholesterol enriches at DMVs assembly sites to maintain membrane fluidity and curvature (18-20)." However, these references refer to hepatitis C virus and enteroviruses. This is just one example and there are several more of that (e.g. on APOE). The authors should be more specific when making such claims to which virus these data refer to.

* The same applies to used cell lines. They need to be properly described throughout the manuscript. For instance, Figure S2, which cell line is this?

* The authors frequently report a "direct" interaction between nsp4 and GRAMD1C. However, the authors cannot exclude the interaction to be indirect, mediated via an adaptor protein. AlphaFold allows to make predictions, but they need to be confirmed experimentally and the pull-down data do not exclude an indirect interaction. Therefore, these statements need to be toned-down, mentioning just "interaction" and omitting "direct".

* Figure 1E, G, I, Figure 5C: Please add quantification of Imaging Data.

The images are very small and thereby hard to evaluate (although the phenotypes displayed are admittedly very clear, but smaller differences as in 5C are not as clear).

* Figure 2D and TEM analysis in other figures: Is it really number of DMV per cell? This would require a complete cross-section of the whole-cell volume; or is it per cell profile of view field? I did not find any statement how the TEM imagens have been quantified. That needs to stated.

* Figure 3: What is the subcellular distribution of GRAMD1C expressed alone (i.e. without TGEV)?

* Figure 3A: What are the co-localization parameters for GRAMD1C and nsp3? By eye it is hard to identify a difference to the GRAMD1C co-localization with nsp4.

* Figure 3B: Was the pulldown performed under expression of nsp4 only or nsp3/4 co-expression? It would be very valuable to validate GRAMD1C preference for nsp4 over nsp3.

* Figure 4: It would be helpful to include replication of naïve/control cells in order to judge the degree of rescue relative to cells without GRAMD1C KO.

* Figure 4B: Why did the authors use an overexpression setting rather than the endogenous protein? That should be explained.

* Figure 4D: The authors use an imaging-based setting of cholesterol and conclude that "… indicating that cholesterol consumption is necessary for productive viral replication." However, this is not obvious from the imaging data. Therefore, a more biochemical approach such as TLC would be more appropriate.

* Figure 4F: Can the authors rescue viral replication in GRAMD1C KO cells by the addition of cholesterol? This would be a most convincing way to demonstrate that cholesterol is responsible for the observed phenotype.

* The model in Figure 5J is nice, but I wonder whether it is correct that the TM of GRAMD1C really spans both ER lipid bilayer and the ER lumen as indicated.

* The discussion could be improved by considering earlier publications on cholesterol transport by OSBP and the entry inhibition by 24HC, and how this relates to the findings by the authors with 20α-HC.

---

## [Decision Letter · Decision Letter 2]

17 Feb 2026

Dear Dr Guiqing,

Thank you for your patience while we considered your revised manuscript "Non-vesicular cholesterol transporter GRAMD1C as a pan-coronavirus antiviral target" for consideration as a Research Article at PLOS Biology. Your revised study has now been evaluated by the PLOS Biology editors, the Academic Editor and the original reviewers.

As you will see in the reports, the reviewers recognize the good job done addressing their previous concerns, but Reviewer #2 still would like some aspects to be addressed. Reviewer #2 still raises several relevant concerns, mainly related to data interpretation and completeness. The reviewer questions the downplaying of a 25% ACE2 reduction and asks that you acknowledge that altered viral entry cannot be excluded, requests inclusion of missing expression data for ANPEP rescue experiments, asks for clarification of the claimed nsp4 specificity given similar correlations with nsp3, and recommends inclusion of additional controls and quantification for newly added experiments. We agree with these concerns and we would require them to be addressed.

In light of the reviews, which you will find at the end of this email, we are pleased to offer you the opportunity to address the remaining points from the reviewers in a revision that we anticipate should not take you very long. We will then assess your revised manuscript and your response to the reviewers' comments with our Academic Editor aiming to avoid further rounds of peer-review, although we might need to consult with the reviewers, depending on the nature of the revisions.

**IMPORTANT - SUBMITTING YOUR REVISION**

*Resubmission Checklist*

*Published Peer Review*

*PLOS Data Policy*

*Blot and Gel Data Policy*

Sincerely,

Melissa

Melissa Vazquez Hernandez, Ph.D.

Associate Editor

PLOS Biology

REVIEWERS' COMMENTS

Reviewer #1:

The authors have satisfactorily addressed my concerns.

Reviewer #2:

In this revision, Su et al have well addressed most of my comments. The authors provide additional data addressing gaps and unclear aspects existing in the first version of the manuscript. Overall, I consider this a greatly improved version of the study that addresses a relevant and interesting aspect of coronavirus biology. Nevertheless, a few minor points need further consideration.

1. Answer to major point 2: "RT-qPCR analysis revealed only a minimal decrease in ACE2 expression, whereas CEACAM1 levels remained unchanged (Fig. S2B-S2D, revised version)": To call a 25% decrease "minimal" is an understatement considering the comparably weak reduction in SARS-CoV-2 replication. The authors still cannot exclude that reduced ACE2 expression contributes to the phenotype. Therefore, the authors should tone down their statement and indicate that while replication appears to be the primary effect of GRAMD1C, reduced viral entry as additional factor responsible for the phenotype cannot be excluded.

2. Regarding the same comment, the authors state "… we rescued ANPEP expression in GRAMD1C-KO cells (GRAMD1C-KO+ ANPEP), restoring ANPEP expression to WT levels." However, this data (ANPEP expression level in the reconstituted cells in comparison to naïve cells) is not shown, but needs to be included.

3. Regarding major point 8: I thank the authors for including this experiment in the manuscript. However, since the Pearson correlation coefficient for both conditions is very high (PCC of >0.7 is considered a strong correlation), bulk analysis of such samples is rarely robust. Therefore, the authors should consider to show exemplary line intensity profiles along a line drawn through the dsRNA puncta and demonstrating that ER signal decreases and cholesterol signal increases in this spot and showing 2 more exemplary cells in the supplement to underline the consistency of this phenotype.

4. Regarding minor points 6 and 7: The correlation coefficient for GRAMD1C with single nsp3 and nsp4, respectively is similarly strong. This needs to be discussed since the authors claim nsp4-specificity, but taken together with their revised figure 3, it appears that nsp3 and 4 are equally responsible for GRAMD1C recruitment, which needs to be clearly stated and discussed.

5. Regarding minor point 12: This is an important result and the authors should consider to include it into the manuscript. However, this would require two positive controls: TGEV in wt cells without GRAMD1C KO +/- cholesterol and proper image quantification.

---

## [Editor Report · Decision Letter 3]

9 Mar 2026

Dear Dr Guiqing,

Thank you for your patience while we considered your revised manuscript "Non-vesicular cholesterol transporter GRAMD1C as a pan-coronavirus antiviral target" for publication as a Research Article at PLOS Biology. This revised version of your manuscript has been evaluated by the PLOS Biology editors and the Academic Editor.

Based on our Academic Editor's assessment of your revision we are likely to accept this manuscript for publication. Please also make sure to address the following data and other policy-related requests.

1) We routinely suggest changes to titles to ensure maximum accessibility for a broad, non-specialist readership, and to ensure they reflect the contents of the paper. In this case, we would suggest a minor edit to the title, as follows. Please ensure you change both the manuscript file and the online submission system, as they need to match for final acceptance:

"The non-vesicular cholesterol transporter GRAMD1C is a pan-coronavirus antiviral target"

2) The Ethics statement needs to be the first subheading in the Material & Methods section. Could you please move it?

3) Please note that per journal policy, the model system/species studied should be clearly stated in the abstract of your manuscript. In this case, please mention all coronavirus tested.

Please supply the numerical values either in the a supplementary file or as a permanent DOI’d deposition for the following figures:

Figure 1D-K, 2B-I, 3I, 4EGH, 5B-K, 6ABCGIJKO, S1ABDFH, S2BCD, S3B, S4AB, S5AB, S6A, S11A-F

→ I am aware you say the raw data has been uploaded to dryad but the link does not work.

5) Please cite the location of the data clearly in all relevant main and supplementary Figure legends, e.g. “The data underlying this Figure can be found in S1 Data” or “The data underlying this Figure can be found in https://doi.org/10.5281/zenodo.XXXXX”

6) Thank you for providing the original, uncropped and minimally adjusted images supporting all blot and gel results reported. However, we wondered if the raw image from Fig S10 is mislabeled as S9.

7) Please ensure that your Data Statement in the submission system accurately describes where your data can be found and is in final format, as it will be published as written there

8) Per journal policy, if you have generated any custom code during the course of this investigation, please make it available without restrictions. Please ensure that the code is sufficiently well documented and reusable, and that your Data Statement in the Editorial Manager submission system accurately describes where your code can be found. More information on our Code Policy, what and how to share can be found here: https://journals.plos.org/plosbiology/s/code-availability

We expect to receive your revised manuscript within two weeks.

*Published Peer Review History*

*Press*

Sincerely,

Melissa

Melissa Vazquez Hernandez, Ph.D.

Associate Editor

PLOS Biology

---

## [Editor Report · Decision Letter 4]

15 Mar 2026

Dear Dr Guiqing,

Thank you for the submission of your revised Research Article "The non-vesicular cholesterol transporter GRAMD1C is a pan-coronavirus antiviral target" for publication in PLOS Biology. On behalf of my colleagues and the Academic Editor, Frank Kirchhoff, I am pleased to say that we can in principle accept your manuscript for publication, provided you address any remaining formatting and reporting issues. These will be detailed in an email you should receive within 2-3 business days from our colleagues in the journal operations team; no action is required from you until then. Please note that we will not be able to formally accept your manuscript and schedule it for publication until you have completed any requested changes.

PRESS

Sincerely,

Melissa

Melissa Vazquez Hernandez, Ph.D., Ph.D.

Associate Editor

PLOS Biology
